



# Fate of sea ice in the 'New Arctic': A database of daily Lagrangian Arctic sea ice parcel drift tracks with coincident ice and atmospheric conditions

Sean Horvath[1,2], Linette Boisvert[1], Chelsea Parker[1,2], Melinda Webster[3], Patrick Taylor[4], Robyn Boeke[5]

[1]NASA Goddard Space Flight Center, 8800 Greenbelt Rd., Greenbelt, MD 20771, USA
[2]Earth System Science Interdisciplinary Center, University of Maryland, 5825 University Research Court Suite 4001, College Park, MD 20740, USA
[3]University of Alaska Fairbanks, Geophysical Institute, 2156 Koyukuk Drive, Fairbanks, AK 99775, USA
[4]NASA Langley Research Center, Climate Science Branch, Hampton, VA 23681, USA
[5]Science Systems Applications Inc., Hampton, VA 23666, USA

*Correspondence to*: Sean Horvath (sean.m.horvath@nasa.gov)

**Abstract.** Since the early 2000s, sea ice has experienced an increased rate of decline in thickness and extent and transitioned to a seasonal ice cover. This shift to thinner, seasonal ice in the 'New Arctic' is accompanied by a reshuffling of energy flows at the surface. Understanding the magnitude and nature of this reshuffling and the feedbacks therein remains limited. A novel database is presented that combines satellite observations, model output, and reanalysis data with daily sea ice parcel drift tracks produced in a Lagrangian framework. This dataset consists of daily time series of sea ice parcel locations, sea ice and snow conditions, and atmospheric states. Building on previous work, this dataset includes remotely sensed radiative and turbulent fluxes from which the surface energy budget can be calculated. Additionally, flags indicate when sea ice parcels travel within cyclones, recording distance and direction from the cyclone center. The database drift track was evaluated by comparison with sea ice mass balance buoys. Results show ice parcels generally remain within 100km of the corresponding buoy, with a mean distance of 82.6km and median distance of 54km. The sea ice mass balance buoys also provide recordings of sea ice thickness, snow depth, and air temperature and pressure which were compared to this database. Ice thickness and snow depth typically are less accurate than air temperature and pressure due to the high spatial variability of the former two quantities when compared to a point measurement. The correlations between the ice parcel and buoy data are high, which highlights the accuracy of this Lagrangian database in capturing the seasonal changes and evolution of sea ice. This database has multiple applications for the scientific community; it can be used to study the processes that influence individual sea ice parcel time series, or to explore generalized summary statistics and trends across the Arctic. Applications such as these may shed light on the atmosphere-snow-sea ice interactions in the changing Arctic environment.



## 1 Introduction

Drastic changes occurring in the Arctic sea ice cover in recent years (Comiso, 2002; Stroeve et al., 2007) have been a topic of great concern not only for the scientific community and local inhabitants, but also for the general public, policymakers and stakeholders. This is because 'what happens in the Arctic does not remain in the Arctic' (Francis & Vavrus, 2012; Vihma, 2014), and changes there will have profound effects politically, economically, ecologically and climatologically on Earth. The Arctic is experiencing the largest temperature increases on our planet (IPCC, 2013) due to global warming. This

process is known as Arctic Amplification (Manabe & Stouffer, 1980; Serreze et al., 2009) and is driving the rapid changes in the Arctic. The most striking change is the decline in Arctic sea ice extent since the late 1970s (Cavalieri & Parkinson, 2012; Parkinson & DiGirolamo, 2016). Since the early 2000s, sea ice has experienced an increased rate of decline in thickness and volume, and transitioned to a predominantly seasonal ice cover (Maslanik et al., 2011; Nghiem et al., 2007) compared to a perennial ice cover in the 1980-1990s (e.g. Comiso et al., 2008; Kwok et al., 2009; Lindsay & Schweiger, 2015). During this

time, observations suggest that the Arctic has become warmer and wetter (Boisvert & Stroeve, 2015), evaporation from the ice-free ocean has increased (Boisvert et al., 2015), the surface albedo has darkened (Duncan et al., 2020), and cloud cover has also increased (Walsh et al., 2011a, 2011b). This era with these large changes observed in the Arctic climate system has been coined the 'New Arctic'.

The shift to thinner, seasonal ice in the 'New Arctic' is accompanied by a reshuffling of energy flows at the surface (Vihma, 2014). Understanding of the magnitude and nature of the reshuffling of the Arctic surface energy budget (SEB) and the feedbacks therein remains limited. This knowledge gap is illustrated by the large spread in projections of the changes in surface turbulent fluxes, the near surface temperatures, and hence lower tropospheric stability (Boeke & Taylor, 2018; Taylor et al., 2018). The temperature structure of the lower atmosphere and changes in the SEB (induced by the changes in

the sea ice pack) are leading drivers of Arctic Amplification (Boeke & Taylor, 2018; Pithan & Mauritsen, 2014). Therefore, synthesizing observations to better understand the evolution of the lower tropospheric temperature structure and its influence on the SEB is critical for improving model predictive capabilities of Arctic Amplification and future sea ice change.

Sea ice growth and melt are driven by changes in the SEB, which is influenced by atmospheric forcing and climate

variability (Holland & Bitz, 2003; Ogi & Wallace, 2012; Taylor et al., 2018). Huang et al. (2019) have recently shown that the accumulation of radiative energy at the surface in early summer (June, July, and August) is a good predictor of September sea ice extent (i.e., sea ice survival). Sea ice thickness has been shown to be important for predicting September sea ice area up to 6 months in advance (e.g., Chevallier & Salas-Mélia, 2012). Thus, understanding what drives the year-to-year variability of sea ice thickness and extent, through winter preconditioning and melt season evolution, can help elucidate

the drivers behind different projected trends in Arctic sea ice loss.



A quantitative understanding of the interaction between sea ice and the atmosphere is important for describing the coupled Arctic climate system and is necessary for improving model physics, which, in turn, can improve seasonal forecasts and climate projections of the fate of the sea ice in the 'New Arctic'. Previous studies have examined and quantified sea ice-
atmospheric interactions using an Eulerian framework (Lynch et al., 2016; Mortin et al., 2016; Stroeve et al., 2016). However, given sea ice mobility, this is a serious limitation to process-oriented understanding by inhibiting the ability to track cumulative effects of atmospheric processes on the SEB and sea-ice mass balance. Therefore, we present the creation of a database to monitor the memory of the sea ice parcels using a Lagrangian framework, tracking their daily motion, characteristics, SEB, and associated atmospheric conditions as they undergo seasonal evolution and drift through the Arctic
Ocean between October 2002-September 2019. The database starts in 2002 as this is highlights conditions in the 'New Arctic' and due to availability of important satellite data. This framework will enable the scientific community to effectively monitor and analyze the evolution of the sea ice and SEB over a variety of atmospheric and sea ice conditions. This effort uniquely unifies a wide variety of satellite and reanalysis data and can provide crucial knowledge of how the 'New Arctic' sea ice couples with the atmosphere, and also how a range of atmospheric forcings and episodic weather events influence the
SEB, sea-ice mass balance, and hence seasonal evolution of Arctic sea ice.

## 2 Data and Methods

### 2.1 Lagrangian Tracked Sea Ice Parcels

Sea ice parcels are tracked in a Lagrangian framework to investigate how sea ice characteristics and their SEB co-evolve and respond to atmospheric conditions. Beginning on 1 October 2002, sea ice parcels are identified in 25-km grid cells where sea
ice concentrations are >15% and given a unique identification number. Adapting the Lagrangian approach in (Tschudi et al., 2010), the location of each sea-ice parcel is tracked daily using the Polar Pathfinder Daily 25-km Equal-Area Scalable Earth (EASE) Grid Sea Ice Motion Vectors (Tschudi et al., 2019). Weekly ice motion data are linearly interpolated to daily vectors. If sea ice concentrations fall below 15%, a parcel's tracking is ceased. If more than 15% sea ice concentration materializes in open water grid cells, a new sea ice parcel is identified and tracked. A separate simulation is run for each
year, running from the beginning of October to the end of September of the following year (i.e. October 2002 - September 2003). Each year, new sea ice parcels are identified at the beginning of October. Sea ice parcels that did not melt out by the end of September are "linked" with the sea ice parcels identified the following October and are flagged as multiyear ice (see "Outputs from Database" section below for more details). At the time of writing, the database includes data through September, 2019.


At each time-step, daily averaged variables of interest are incorporated as individual data layers for each sea ice parcel in three separate categories: 1) sea ice conditions, 2) SEB terms between the parcel and atmosphere, and 3) atmospheric conditions (see Table 1 and Figure 1). All variables are re-projected to the EASE projection to match the coordinate



reference system projection of the sea ice trajectories (Brodzik et al., 2012; Brodzik & Knowles, 2002). Along with these
data layers, flags are given to denote the presence/absence of episodic weather events. The flag is determined based on the
co-location of the ice parcel and independently tracked cyclones. If a cyclone is present at the location of the parcel, the
parcel is flagged with individual system identifications along with distance from the center of the system, cardinal direction
from the center of the system, and system minimum surface pressure as a measure of intensity. By incorporating synoptic
event information, users are able to pinpoint perturbations that may result in propagating effects on the SEB and sea ice mass
balance. Currently only cyclones are flagged, however future work plans to include other types of episodic weather events.

## 2.2 Sea Ice Conditions

### 2.2.1 Sea Ice Concentration

Two sea ice concentration data products are included in this database, the Sea Ice Concentration Climate Data Record (CDR)
(Meier et al., 2021, https://nsidc.org/data/G02202/versions/4| National Snow and Ice Data Center) and the Sea Ice Index
product     housed     at     the     National     Snow     and     Ice     Data     Center     (NSIDC)     (Fetterer     et     al.,     2017,
https://nsidc.org/data/G02135/versions/3). Both datasets are derived from two sources: (1) the Near-Real-Time Daily Polar
Gridded Sea Ice Concentrations (NRTSI) from the Special Sensor Microwave Imager/Sounder (SSMI/S) on board the
Defense Meteorological Satellite Program (DMSP) satellites (Maslanik & Stroeve, 1999) and (2) the DMSP Special Sensor
Microwave/Imager (SSM/I, 1987–2007), and the Special Sensor Microwave Imager/Sounder (SSMI/S, 2007 to 2019). The
Sea Ice Index uses the NASA Team Algorithm (Cavalieri et al., 1984) for sea ice concentration estimates, while the CDR is
a rule-based combination of the NASA Team Algorithm and the NASA Bootstrap algorithm (Comiso, 1986). Sea ice
concentrations derived from the NASA Team algorithm are used for the Lagrangian tracking method described above.
Multiple sources of sea ice concentration are included as different algorithms perform better than others in certain conditions
(i.e. low concentrations), although the trends in sea ice area and extent tend to agree (Ivanova et al., 2014)

### 2.2.2 Sea Ice Thickness

Continuous, daily sea ice thickness estimates from observations are lacking over this time period. Therefore, daily sea ice
thickness is obtained from the Pan-Arctic Ice-Ocean Modeling and Assimilation System (PIOMAS), a coupled ocean and
sea ice model that focuses on the Arctic ocean (Zhang & Rothrock, 2003, http://psc.apl.uw.edu/research/projects/arctic-sea-
ice-volume-anomaly/data/model_grid).      PIOMAS     is     driven     by     National     Centers     for     Environmental     Prediction
(NCEP)/National Center for Atmospheric Research (NCAR) reanalysis data and is formulated in a generalized orthogonal
curvilinear coordinate system which is used for bilinear interpolation with every sea ice parcel location. This grid has a mean
horizontal resolution of 22km for the Arctic Ocean. Sea ice thickness values produced by PIOMAS compare favorably with
ICESat measurements, with a correlation of 0.83 and root mean squared error of 0.61m for spring (February-March), and a
correlation of 0.65 and root mean squared error of 0.76m for autumn (October-November) (Schweiger et al., 2011).



### 2.2.3 Snow Depth and Density

As with sea ice thickness, continuous, daily snow depth estimates from observations are not available for incorporation into this product. Daily, pan-Arctic snow depth and density on a 25-km × 25-km grid are obtained from the Lagrangian snow-evolution model (SnowModel-LG) (Liston et al., 2020, https://nsidc.org/data/NSIDC-0758/versions/1). The model is forced with NASA's Modern Era Retrospective-Analysis for Research and Applications-Version 2 (MERRA-2) and European Centre for Medium-Range Weather Forecasts (ECMWF) ReAnalysis-5th Generation (ERA5) atmospheric reanalysis products, providing two individual sets of snow properties. By performing full surface and internal energy balances and mass balances within a multilayer snowpack evolution system, SnowModel-LG accounts for rainfall, snowfall, sublimation from static-surfaces and blowing-snow, snow melt, snow density evolution, snow temperature profiles, energy and mass transfers within the snowpack, superimposed ice, and ice dynamics. The redistribution of snow particles due to wind is not included in SnowModel-LG as the sea ice parcel sizes (14 x 14 km in SnowModel-LG) are too large to simulate snow erosion and deposition. Other possibly important processes that are not incorporated in SnowModel-LG include snow blowing into leads and snow-ice formation (when seawater floods the snowpack and refreezes due to a heavy snow load that submerges the ice surface below sea level). SnowModel-LG outputs have shown reasonable agreement with ice mass balance (IMB) buoys and measurements from the Surface Heat Budget of the Arctic Ocean (SHEBA) experiment and the Norwegian young sea ICE (N-ICE2015) measurements (J. Stroeve et al., 2020).

## 2.3 Atmospheric Conditions

### 2.3.1 ERA5

Atmospheric values are from ECMWF Reanalysis 5th Generation (ERA5) (Hersbach et al., 2018, https://www.ecmwf.int/en/forecasts/dataset/ecmwf-reanalysis-v5). ERA5 was produced using 4D-Var data assimilation in CY41R2 of ECMWF's Integrated Forecast System (IFS), with 137 hybrid sigma/pressure (model) levels in the vertical, with the top level at 0.01 hPa. Values have a spatial resolution of 0.25° latitude by 0.25° longitude. Surface values used here include total precipitation, snowfall, skin temperature, surface pressure, 2-meter air temperature, total column water vapor, and 10-meter wind speed and direction. Specific humidity, air temperature, and wind speed and direction at four pressure levels (1000hPa, 925hPa, 850hPa, 500hPa) are also available. Although atmospheric data from ERA5 correlates well with in situ measurements taken during the N-ICE2015 campaign, ERA5 was found to have a large positive bias in 2m temperature in winter and spring (Graham et al., 2019).

### 2.3.2 MERRA-2

Atmospheric values are derived from the Modern Era Retrospective Analysis for Research and Applications (MERRA-2) (Gelaro et al., 2017, https://gmao.gsfc.nasa.gov/reanalysis/MERRA-2/data_access/). MERRA-2 uses the Goddard Earth Observing System, Version 5.12.4 (GEOS-5) atmospheric model and Global Statistical Interpolation (GSI) analysis scheme

and has an approximate spatial resolution of 0.5° latitude by 0.625° longitude. Surface values used here include total precipitation, snowfall, skin temperature, 2-meter air temperature, surface pressure, total column water vapor, 10-meter wind speed and direction, total precipitable water, and total precipitable snow. Specific humidity and air temperature at two pressure levels (850hPa & 500hPa) and wind speed and direction at three pressure levels (850hPa, 500hPa, & 250hPa) are

also available. Although atmospheric variables from MERRA-2 correlate well with in situ measurements during the N-ICE2015 campaign, MERRA-2 was found to have large biases in the total column water vapor in spring and summer (Graham et al., 2019).

Inclusion of both MERRA-2 and ERA5 variables provides flexibility for potential applications. For instance, SnowModel-

LG produces two sets of snow characteristics, one forced with MERRA-2 and one forced with ERA5. Inclusion of atmospheric variables from each reanalysis model enables matching snow-atmosphere comparisons.

### 2.3.3 AIRS

NASA's Atmospheric Infrared Sounder (AIRS) onboard the Aqua satellite was launched in May 2002 and has been collecting twice daily, global data ever since. AIRS has 2378 infrared channels and a 13.5 km spatial resolution. The AIRS

instrument was designed to produce highly accurate temperature and humidity profiles globally (Susskind et al., 2014), which is important in the Arctic where data are sparse and clouds are prevalent. AIRS Version 6 temperatures and humidity products have been compared with a variety of in-situ data and have shown to have modest errors in skin temperature (2.3K), 2m air temperature (3.41K) and specific humidity (0.55 g/kg) (Boisvert et al., 2015; Taylor et al., 2018). V7, AIRS-only atmospheric variables are used and include single level values of skin temperature, surface air temperature, and total column

precipitable water as well as air temperature, geopotential height, and relative and specific humidity at 6 pressure levels (1000hPa, 925hPa, 850hPa, 700hPa, 600hPa, and 500hPa). Some of these variables are also provided by MERRA-2 and ERA5, but AIRS provides an observational perspective to complement the model derived variables.

### 2.3.4 Cyclones

The Melbourne University cyclone tracking scheme (Simmonds et al., 2008) is used for identifying closed cyclone systems

due to its consistency in capturing cyclone events, its broad agreement in results with other cyclone tracking algorithms (Neu et al., 2013; Raible et al., 2008), and the availability of methodology from Webster et al., (2019). To describe the tracker briefly, sea level pressure (SLP) fields from ERA5 reanalysis are regridded and smoothed to 1-degree resolution. The Laplacian (LP) of the SLP fields are then calculated to determine the local maxima of LP relative to eight neighboring grid cells. Once these local maxima are identified, a set of criteria are imposed: (a) the second derivative of the SLP in the x- and

y-directions must be positive, and (b) the mean LP in the immediate vicinity of the maxima must meet the "concavity criterion" where LP is equal to or greater than 0.2 hPa per degree latitude squared. At every 6-hourly time-step, the cyclone centers are determined through an iterative approach that finds the minimum first derivatives (in x and y) within the local



area of a center candidate, identifying both open and closed systems. The cyclone area is determined by fitting an ellipse to the near-zero LP values in eight opposing directions from the cyclone center. For the purpose of the ice parcel database, only closed systems are included and given unique identifiers representing individual systems. All points within the cyclone area are flagged as the same cyclone event, and if multiple cyclones overlap in a given area, those points are flagged with each cyclone ID.

## 2.4 Surface Energy Budget

### 2.4.1 Clouds and Earth's Radiant Energy System (CERES)

The CERES instrument is used by the Radiation Budget Science Project at NASA Langley to produce surface, atmosphere, and top-of-atmosphere radiative fluxes. These data products range from the instantaneous fluxes for each ~20km CERES footprint to monthly, gridded radiative fluxes. This product incorporates CERES radiances, Moderate Resolution Imaging Spectroradiometer (MODIS) cloud properties, surface albedo retrievals, and meteorological information from the Global Modeling and Assimilation Office (GMAO) to produce 1-hourly resolved surface, atmosphere, and top-of-atmosphere radiative fluxes. Comparisons between the CERES longwave (LW) and shortwave (SW) surface fluxes and surface radiometer observations show uncertainties ~6% in the longwave and 23% in the shortwave at the hourly, regional time scale over global ocean and land (Kato et al., 2013). Clear-sky/all-sky surface and top-of-atmosphere radiative fluxes are obtained from the CERES CERES-SYN1DEG product (NASA/LARC/SD/ASDC, 2017; Wielicki et al., 1996; https://ceres.larc.nasa.gov/data/#syn1deg-level-3).

### 2.4.2 Atmospheric Infrared Sounder (AIRS)

The turbulent flux terms of sensible and latent heat are produced using AIRS Version 7 Level 3 data products of skin temperature, 925 and 1000 hPa air temperature, relative humidity, and geopotential height (https://airs.jpl.nasa.gov/data/get-data/standard-data/), MERRA-2 10m wind speed and passive microwave sea ice concentration produced using the NASA Team algorithm (Cavalieri et al., 1996, updated yearly). Turbulent fluxes are estimated using the bulk aerodynamic method with the Monin-Obukhov Similarity Theory and an iterative calculation based on Launiainen & Vihma (1990) on the $25\text{km}^2$ polar stereographic grid. These fluxes are derived with a few modifications that were tailored specifically to capture the unique conditions of the boundary layer and roughness of the Arctic sea ice (see Boisvert et al., 2013; 2015 for more information). These Arctic sea ice specific changes made to this algorithm have not been adapted or included in any other climate models or reanalysis products and are better suited to simulate turbulent fluxes from the Arctic Ocean. In fact, when compared with in situ data from the Norwegian Young Sea Ice Experiment (N-ICE2015) campaign, AIRS latent and sensible heat fluxes had errors of 0.74 W/m2 and 5.32 W/m2, respectively (Taylor et al., 2018). Overall, these comparisons produce an error of ~20% in the AIRS-derived surface turbulent fluxes, but provide the most complete picture of Arctic surface turbulent fluxes over a 20-year period, in the absence of in situ data.




The database presented in this work is the first to incorporate the CERES surface radiative fluxes (NASA/LARC/SD/ASDC, 2017) and AIRS surface turbulent flux data. This enables the complete characterization of the SEB evolution of sea ice parcels across the Arctic domain.

## 2.5 Validation of Remote Observations

Data from sea ice mass balance (IMB) buoys is obtained from the CRREL-Dartmouth Mass Balance Buoy Program
(Perovich et al., n.d.). These buoys were deployed in various regions throughout the Arctic ocean and recorded sea ice thickness, snow depth, and air temperature and pressure along with GPS locations. IMB buoys provide 4-hourly data that are aggregated here to daily means. The ice mass balance buoy data are incorporated into the ice motion data (Tschudi et al., 2019) of the ice parcel database and do not provide independent validation of drift location. However, the drift location of the buoys are useful for evaluating the derived Langragian sea ice parcel tracks. Additionally, comparing sea ice parcels with
these buoys does provide independent validation for time series of key sea ice/snow/atmospheric variables.

## 3 Results

### 3.1 Outputs from Database

The database is stored in HDF5 file format where an individual file exists for each unique sea ice parcel (Figure 2). The top level contains the location of the sea ice parcel at daily time steps along with metadata. A group exists for each data source
that has been combined with the sea ice parcel (AIRS, PIOMAS, MERRA-2, Cyclones, etc.). Every group contains a dataset for each individual variable (split further by pressure level when available). The files are grouped by year ("year" here is from October to the following September) and for each year a metadata file exists summarizing key characteristics of each sea ice parcel which can be used to filter sea ice parcels that meet certain criteria. Because the tracking algorithm is restarted each year at the beginning of October, parcel ID numbers do not carry over from September into October. Instead, an
additional metadata file is included linking all parcels that survived summer melt (e.g., did not melt out by the end of September) with the nearest parcel at the beginning of the following October.

### 3.2 Validation and Assessment of Remote Observations and Simulated Processes

### 3.2.1 Trajectory Evaluation

The modeled sea ice parcel trajectories are assessed by comparing them to sea ice mass balance buoys that have been
deployed throughout the Arctic Ocean (Perovich et al., n.d.). Although IMB buoys are not an independent validation of the sea ice parcel drift locations, they are a useful tool for evaluating the results of the Lagrangian methodology. After removing buoys that do not match the time frame of our database and those that contain erroneous location data, we are left with 74

buoys that are used for comparison. Figure 3a shows the tracks of five buoys (blue) and the track of the closest ice parcel at the time of the buoy deployment (green). There are differences in daily positions, but overall the derived trajectories closely

match the buoy tracks. Sea ice parcels generally remain within 100km of the corresponding buoy (Figure 3b), with distances often much shorter (mean: 83km, median: 54km).

The buoy/parcel pairs originating in the Central Arctic and drifting south through the Fram Strait show the largest discrepancy in ending locations, with the buoy traveling further south than the sea ice parcel (the date of the end point

locations are the same for buoys and sea ice parcels as buoy drifts tend to last longer than that of the ice parcels). While many trajectories remain within 100km of the associated buoy (Figure 3b), those buoys deployed in the Central Arctic occasionally gain greater separation from the ice parcel, upwards of ~1,500km. The buoy/parcel pairs that have distances greater than 500km between them (Figure 4) are trajectories that begin in the Atlantic sector of the Central Arctic and drift south through the Fram Strait (only buoy locations with valid ice thickness measurements are shown as buoys can sometimes

float in open water after the ice pack has melted). The buoys drift faster and further south along Greenland's east coast than the simulated ice parcels, indicating that the ice motion vectors in Fram Strait do not capture the true ice velocities (Sumata et al., 2014).

### 3.2.2 Mass Balance and Atmosphere Comparison

Comparisons of the properties observed by sea ice mass balance buoys and those derived from the sea ice parcel Lagrangian

framework are shown in Figure 5. Sea ice thickness and snow depth show the greatest variability compared to buoys (Figure 5a). This can in part be explained by the spatial variability of these variables within each 25km by 25km grid cell. Because PIOMAS and SnowModel-LG provide averaged values for each grid cell, discrepancies between these values and in situ point sources (buoys) are expected. Using NASA's Operation IceBridge (MacGregor et al., 2021) as a reference, the standard deviation of sea ice thickness within a 25km by 25km grid cell on a given day ranges from 0.01-5.67m, with a mean

of 1.44m and median of 1.43m (data obtained from NSIDC's IceBridge L4 Sea Ice Freeboard, Snow Depth, and Thickness, Version 1; Kurtz et al., 2015). Smaller differences in air temperature and pressure are expected as these values have less spatial variability.

Comparing the sea ice parcel values with buoy data (Figure 5b) provides an evaluation of the data set utility for addressing

science related to the sea ice parcel evolution. All values show high correlation overall with a greater spread for sea ice thickness and snow depth. The mean correlation coefficients are 0.53 for sea ice thickness, 0.56 for snow depth, 0.95 for air temperature, and 0.96 for air pressure. Although there are some negative correlations for sea ice thickness and snow depth for individual parcels, collectively the sea ice parcels largely capture the evolution of the ice pack and are therefore a good source of information on assessing the evolution of the sea ice and snow pack along their drift trajectories annually.



### 3.3 Database Uses

#### 3.3.1 Climatological Studies

In addition to tracking the location and atmospheric - sea ice interactions for individual parcels, this database can be used to assess characteristics of sea ice parcels collectively. With declining sea ice cover in recent years, a reasonable expectation would be a decreasing number of sea ice parcels as well. However, the total number of sea ice parcels per year is increasing at a rate of 365.5 parcels or 228,437 km$^2$ per year (Figure 6a, blue). At the same time, the average duration (in days) of individual sea ice parcels is decreasing at a rate of -1.2 days per year (Figure 6a, green). This demonstrates that the Arctic sea ice cover is transitioning to a more seasonal state, accounting for both the shorter duration of ice parcels and the increase in the total number of unique ice parcels. This result is consistent with the observed transition towards a seasonal sea ice dominated Arctic (Kwok, 2018).

To examine ice parcel freezing/melting events further, Figure 6b shows the total count of parcel generation (freezing, blue) and extinguishing (melting, red) events by month and year. With the exception of freezing in October and melting in April, all trends are positive which can in part explain the simultaneous increase in sea ice parcels and decrease in sea ice parcel duration. Melting trends of 96.5 parcels (~60,312 km$^2$) per year and 153 parcels (~95,625 km$^2$) per year in May and June, respectively, are indicative of an earlier open water season in recent years (Bliss et al., 2019). Similarly, a negative trend in freezing sea ice parcels in October along with increasing trends in freezing in November (-50.4 (-31,500 km$^2$) and 92.8 (58,000 km$^2$) parcels per year, respectively) are representative of a later end to the open water season (Stroeve et al., 2014).

With the shift from a predominantly multiyear ice (MYI) pack to a predominantly first year ice (FYI) pack in the 'New Arctic', the survivability of each of these ice classifications is of keen interest and can be observed as in Figure 7a. The majority of FYI melts out every year while the majority of MYI survives the summer melt season. The interannual variability suggests this database can be used for case studies of particular sea ice years, such as the record low September 2012 extent, where there was a decrease in FYI and MYI that survived the summer melt.

The inclusion of CERES and AIRS data with these sea ice parcel trajectories provides opportunities to examine connections between the SEB and the fate of sea ice. As mentioned earlier, studies have shown that the SEB in June, July, and August can be a good predictor of September sea ice extent (Huang et al., 2019; Sedlar, 2018). The SEB is calculated primarily with NASA remotely sensed observations of the radiative component from CERES and the turbulent flux component derived from AIRS:

$$Fr + FL + FE + FS + Fe = SEB \tag{1}$$

Where, Fr is the net absorbed SW flux, FL is the downwelling LW flux, FE is the upwelling LW flux, FS is the sensible heat
flux and Fe is the latent heat flux. The conductive flux from the ocean through the sea ice is omitted here due to the lack of
observational data.  Figure 7b shows the daily averaged SEB for all sea ice parcels for these summer months, split by
whether the sea ice parcel melted out (red) or survived the melt season (blue).  In each region, the SEB was greater for sea
ice parcels that melted out on average than those that survived. The greatest differences in SEB occurred between days 175
and 200 (late June through mid July) which coincides with peak insolation.

### 3.3.2 Case Studies

An example of tracks and time-series data from the 2017-2018 simulation is shown in Figure 8, where major patterns of sea
ice drift are observable in the sample parcels.  For example, the influence of the Beaufort Gyre can be seen in the light and
dark green trajectories, while the Transpolar Drift can be observed in the blue trajectory.

Figure 9 displays the time-series of select variables relating to the SEB for the light green trajectory seen in Figure 8 that is
advected from the northern Chukchi Sea into the East Siberian Sea. At the beginning of the second week in June (vertical
dotted red line), after about a two week period of consistently high downwelling longwave radiation, the skin temperature
rises above the melting point and corresponds with a decrease in snow depth and albedo, and an increase in snow density.
This could be an indicator of a melt onset event and the beginning of the melt season. Similar quick comparisons like this
can easily be performed using this database of sea ice parcels and corresponding atmospheric conditions from October 2002
to September 2018.

Cyclone flags can be a useful tool for analysis of snow depth changes, precipitation events, and changes to sea ice
concentration induced by sea ice thermodynamics.  Figure 10 shows a sample time series from the CRREL-Dartmouth Mass
Balance Buoy Program (Perovich et al., n.d.) (buoy ID 2004D) and the nearest sea ice parcel at time of deployment (sea ice
parcel ID 2003-2004_19893) where vertical lines indicate the presence of cyclones. As buoys/sea ice parcels can be
influenced by multiple cyclones on the same day, only the nearest cyclone (distance from buoy/ice parcel to cyclone center)
is shown. Because the buoy and sea ice parcel locations differ slightly they often experience different cyclone events (in
Figure 10a, blue/green circles & triangles show characteristics of the nearest cyclone to the buoy/ice parcel). Cyclones with
large daily snowfall in spring after early June coincide with a slow down of the generally decreasing snow thickness, while
cyclones precipitating snowfall in the autumn are followed by a sharp increase in snow depth (Figure 10a). Sea ice
concentration tends to fluctuate regardless of whether a cyclone is in the vicinity, but rarely stays unchanged when a cyclone
is present (Figure 10b). Further analysis can examine relationships between changes to sea ice snow depth/density and sea
ice concentration and the distance and direction of cyclones from sea ice parcels, and whether these relationships depend on
the time of year.



### 3.4 Future Additions

Current work is underway to include additional data layers to the Lagrangian ice parcel database for enabling studies of atmosphere-ice interactions; these data layers include: a daily surface melt and freeze product, indicators for extreme moisture and warm/cold air intrusions, and flags for polar low events. Additional data sets may be added in the future where applicable, and the database will be updated yearly with current data.

### 3.4.1 Daily Melt and Freeze Events

The timing of melt onset and freeze up throughout the melt season is important for sea ice survivability throughout the year, and also in understanding the Arctic climate system (Bliss et al., 2017, 2019; Boisvert & Stroeve, 2015; Mortin et al., 2016; Stroeve et al., 2014). Once surface melt occurs, it does not always mean that continuous melt is occurring. In fact, throughout the summer melt season, the temperatures tend to oscillate around the freezing point and there are multiple melt and refreezing events occurring at the surface (Persson, 2012). A new daily melt onset data product is currently under development. This product builds upon the methodology of Markus et al. (2009) using passive microwave brightness temperatures to measure the emissivity of the surface, and combines sea ice concentrations, and skin and air temperatures from AIRS to determine if a melt or freeze event is occurring on the surface. This data can be used to assess the amount of melt/freeze events that are occurring throughout the evolution of individual sea ice parcels along with atmospheric and snow/sea ice conditions. Once this product is developed, future plans will include applying daily melt/freeze flags to the sea ice parcel Lagrangian database.

### 3.4.2 Episodic Weather Events

Future work will involve creating a database of extreme moisture and temperature intrusions and polar lows, and then applying these flags to the sea ice parcel Lagrangian database using a similar methodology to that used for cyclone events laid out in this paper. Extreme temperature and moisture intrusions will be identified using ERA5 and MERRA-2 atmospheric variables. To identify these events, a modified version of the methodology in Woods et al. (2013) and Woods & Caballero (2016) will be developed. The primary criteria for defining a moisture intrusion is based upon the magnitude, duration, and longitudinal extent of events where: 1) events must be in the top 20% of the total mass-weighted column tropospheric moisture transport across the 70° N polar cap boundary; 2) events must last at least 1.5 days; and 3) zonal extent of events must be at least 9 degrees of longitude. However, the criterion requiring that moisture intrusions must cross the 80° N latitude circle can be relaxed (Hegyi & Taylor, 2018). Unlike in Woods & Caballero (2016), all moisture intrusions that enter the Arctic (crossing 70° N) are of interest as that energy can have significant impacts on the Arctic surface energy budget of the tracked sea ice parcels. The magnitude threshold will also be different than Woods et al. (2013), where the top 10% of moisture transport events into the Arctic was used as a single threshold. Using a lower magnitude threshold (top

20%) allows for the investigation of the differences between weak versus strong events, where the cloud response is expected to be particularly sensitive. This same criterion will also be used to identify warm and cold air outbreaks. Similar to the cyclone database, intrusion flags will be applied to sea ice parcels when they come into contact with one of these events.

A database of polar lows will be created using ERA5 data and an adaptation of the cyclone tracking algorithm (Simmonds et al., 2008; Webster et al., 2019). Polar lows are intense, mesoscale cyclones that are associated with fast propagation speeds; strong winds; high intensity precipitation as snowfall, hail, and/or rainfall; high waves; and freezing sea-spray (Iversen, 2013; Rasmussen & Turner, 2003). Interactions between polar lows and the sea ice / ocean surface remain poorly understood and these events may have important implications for sea ice mass balance. The events will be identified based on criteria of

1) genesis over ocean regions; 2) occurrence between October - May; 3) radii of ~100-500km; 4) presence of a warm core. Similar to the cyclone database, polar low flags will be added as a data layer to the sea ice parcels if they coincide with an event.

### 3.4.3 Multidisciplinary drifting Observatory for the Study of Arctic Climate (MOSAiC)

Once data from the he Multidisciplinary drifting Observatory for the Study of Arctic Climate (MOSAiC) expedition has

been quality checked and processed, these MOSAiC datasets present a unique opportunity to use in situ Lagrangian data to validate and interpret the snow depth and sea ice thickness results from the remotely-sensed ice parcel database presented here.MOSAiC was a year-long field campaign in the central Arctic where the R/V Polarstern was frozen into the pack ice (Shupe et al., 2020). The overarching objective of the MOSAiC expedition was to collect process-oriented, continuous field observations of the Arctic climate system year-round to advance understanding centered on Arctic system science in the

'New Arctic'. The field experiments encompassed nested spatial scales up to 50 km and continuously drifted with the wind and ocean currents between Oct. 2019 – Oct. 2020.

### 4 Conclusion

A 'New Arctic' sea ice parcel database has been presented which combines satellite observations and reanalysis data with daily sea ice parcel drift tracks produced in a Lagrangian framework. This novel dataset contains daily sea ice parcel

locations, sea ice and snow conditions, and atmospheric states and fluxes from 2002-2019. Building on previous work (e.g. Tooth & Tschudi, 2017), this dataset includes drivers of surface energy fluxes from which the SEB can be calculated. Additionally, flags have been included to identify when sea ice parcels are potentially influenced by synoptic events such as cyclones. The dataset records distance and direction from the center of the system as well as cyclone intensity. This dataset allows users to track the movement and evolution of sea ice parcels and the associated atmospheric state as they advect

throughout the Arctic Ocean.

The database trajectories and sea ice and atmospheric characteristics have been compared to data collected from sea ice mass balance buoys from the CRREL-Dartmouth Mass Balance Buoy Program (Perovich et al., n.d.) by identifying the closest ice parcel to the deployment location.  Results show ice parcels generally remain within 100km of the corresponding buoy

(Figure 3b). The largest discrepancies are found near the Fram Strait where observed sea ice velocities tend to be much higher. Compared to the IMB data, the ice parcel ice thickness and snow depth typically are less accurate than air temperature and pressure; this may be attributed to high spatial variability of the former two quantities when compared to a point measurement from a buoy.  The overall high correlation coefficients between buoys and sea ice parcels show that changes in these quantities over time are in good agreement, suggesting the ice parcel database is useful for assessing sea ice

evolution.

General characteristics of collections of sea ice parcels can be determined with this database in addition to analyses of individual sea ice parcel time series. In recent years there has been an increase in the number of sea ice parcels at a rate of 365.5 parcels per year and a decrease in average duration of sea ice parcels at a rate of -1.2 days per year suggesting more

melt and freeze events (i.e., extinguishing and generation of sea ice parcels, respectively) and a transition to a seasonal sea ice cover. The survivability of sea ice parcels are linked to the June, July, and August summed SEB, where parcels that experience a larger flux of energy at the surface are less likely to survive the summer melt season. This can be expanded upon in future work by exploring the impact of individual components contributing to the SEB and the influence of SEB on autumn ice growth.


The results shown here are just the 'tip of the iceberg' in the amount of new research and scientific results that this database will enable. This database has vast applications for the wider scientific community to utilize and to better understand sea ice-atmospheric interactions in the 'New Arctic' and to explore what atmospheric factors and their timing might hinder or aid in the survivability of sea ice throughout the year. It also enables process oriented research when compared to previous eulerian

based investigations. This database could also be used to assess climate model simulations of Arctic variables and processes in order to evaluate and improve model physics. Currently, this dataset is expected to be hosted by the National Snow and Ice Data Center, where it will be available for public download.

Acknowledgements
The work of S. Horvath, L. Boisvert, C. Parker, M. Webster, P. Taylor, and R. Boeke is funded by NASA's Interdisciplinary Science project entitled 'Investigating the fate of sea ice and its interaction with the atmosphere in the New Arctic', grant number 80NSSC21K0264.



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





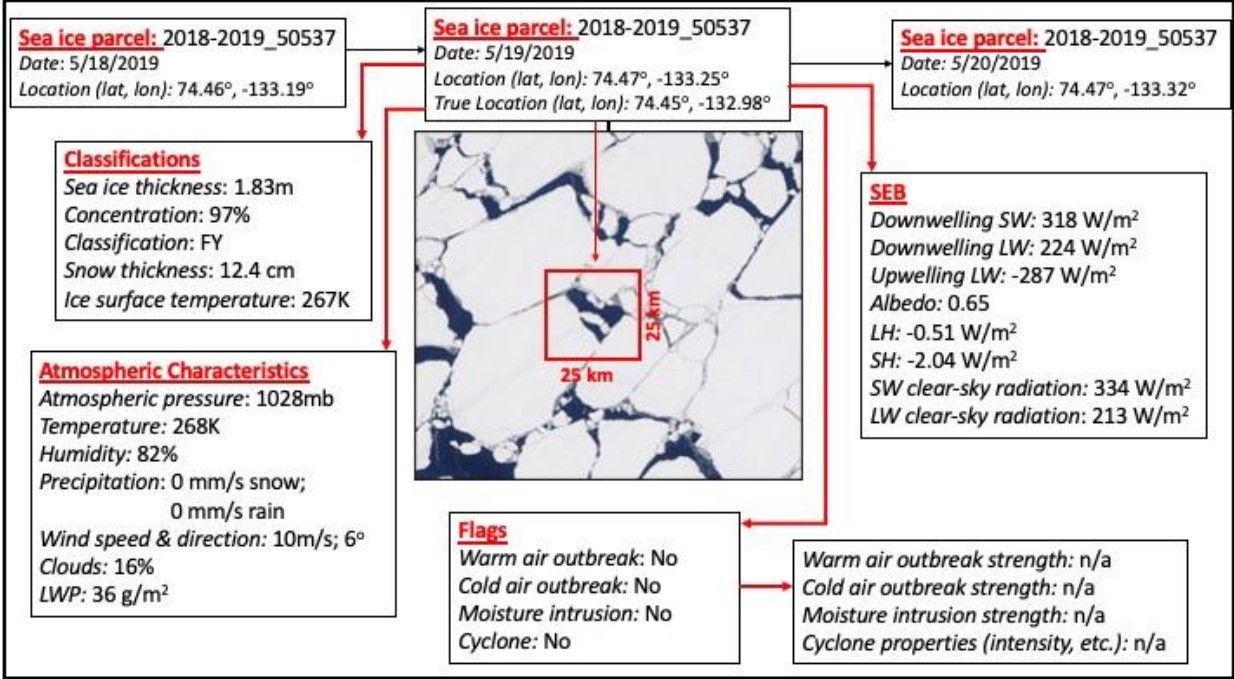


**Figure 1. A schematic and details of the information associated with the Lagrangian tracked sea ice parcel #2018-2019_50537. Each sea ice parcel, on each day of the year, contains sea ice characteristics, atmospheric characteristics, SEB, and episodic weather cyclone event flags at a specific date and location. In the schematic, MY is multi-year sea ice, FY is first-year sea ice, LWP is liquid water path, SW is shortwave radiation, LW is longwave radiation, and LH is latent heat flux and SH is sensible heat flux.**

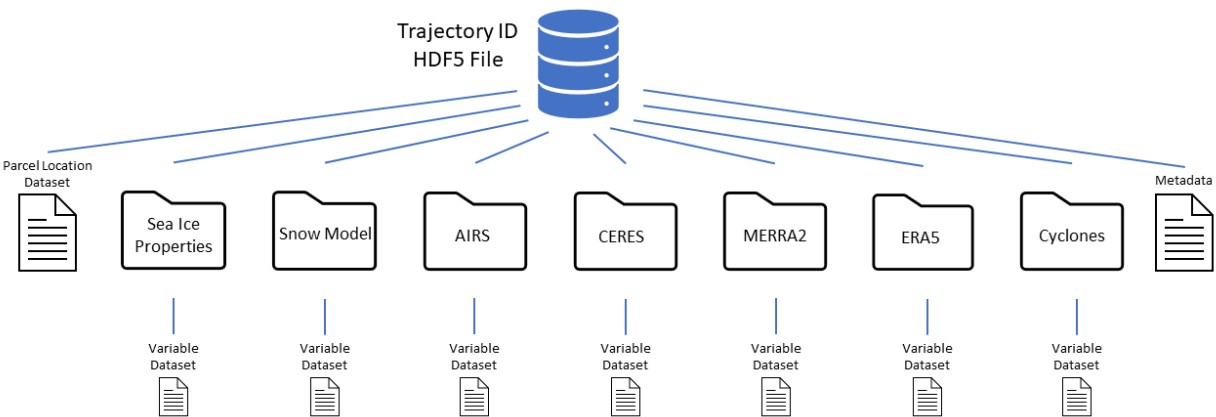


**Figure 2. File structure for individual trajectories in the database.**





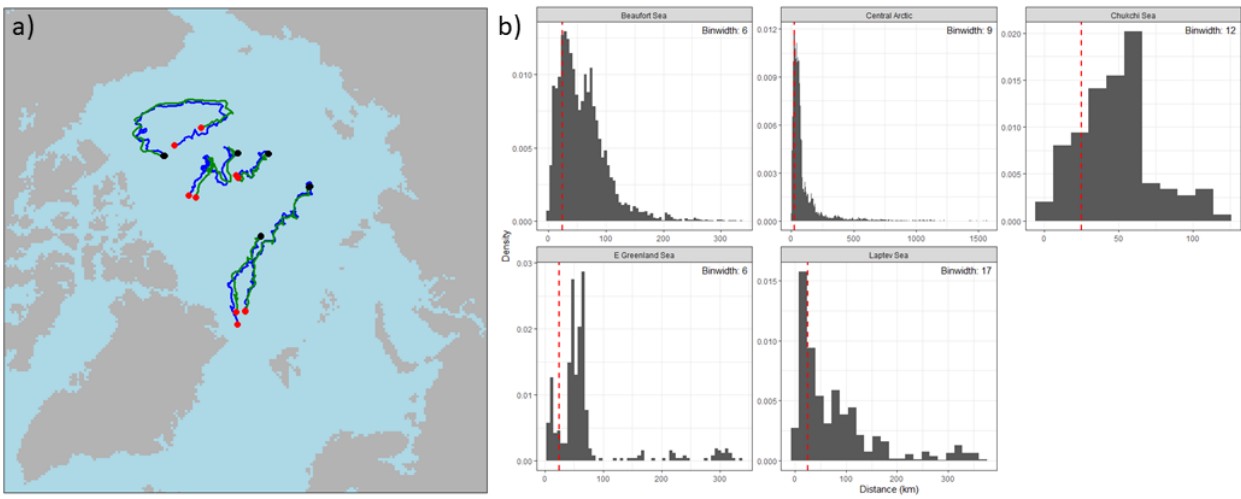

**Figure 3. Comparison of ice mass balance buoy tracks and simulated sea ice parcel tracks. a) Sample of buoy tracks (blue) and the closest ice parcel at time of deployment (green). The black/red dots represent beginning/ending locations. b) Histogram of the daily distance between buoys and ice parcel track by deployment region. Vertical dotted red lines indicate 25km. Binwidth calculated using Freedman-Diaconis rule.**

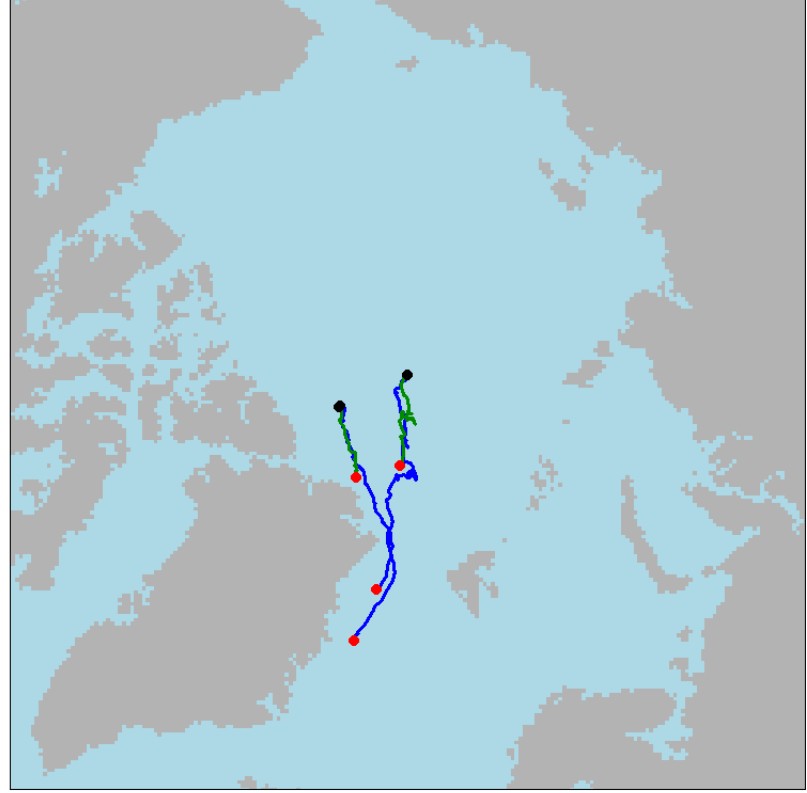

**Figure 4. Buoy tracks (blue) and corresponding sea ice parcel trajectories (green) where the distance between tracks exceeds 500km. Black/red dots indicate beginning/ending locations.**





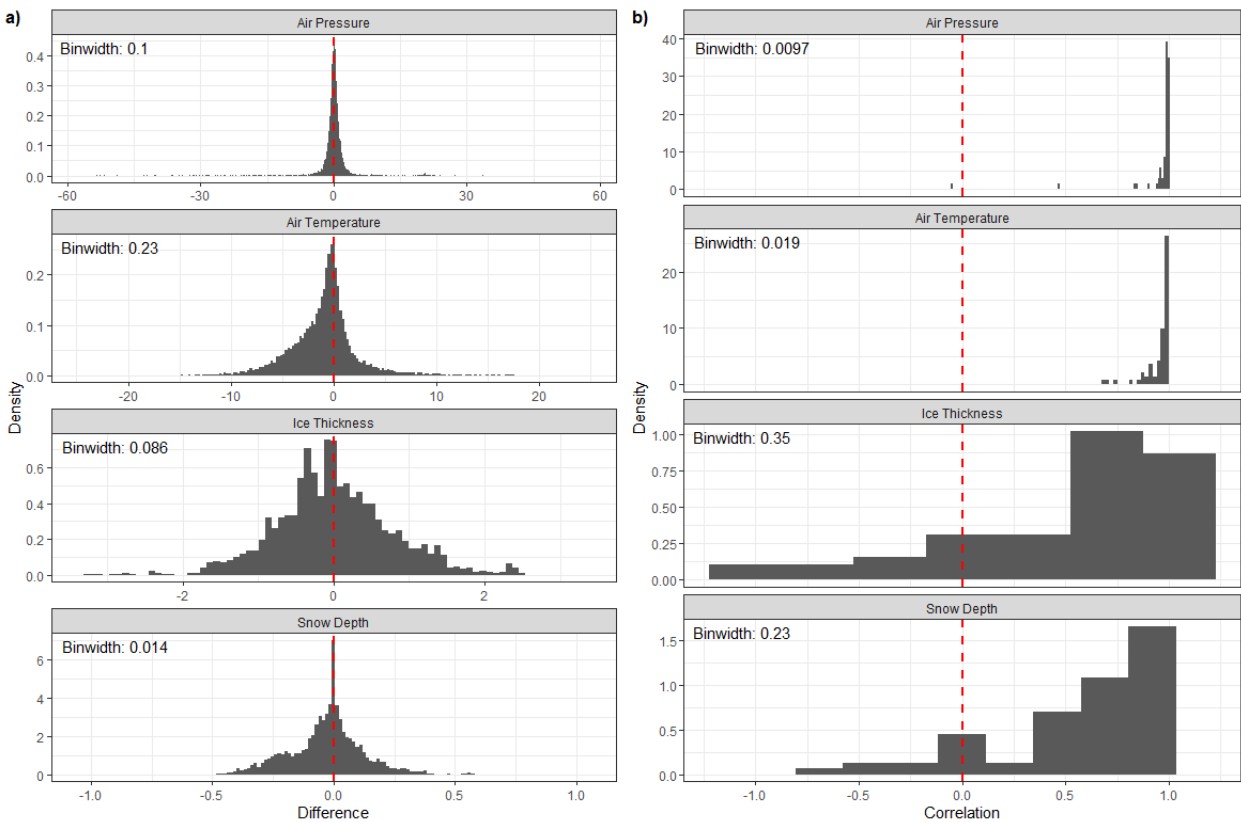


**Figure 5. Comparison between buoys and tracked ice parcels of ice thickness, snow depth (from SnowModel-LG forced with MERRA-2), and air temperature and pressure (parcel air temperature and pressure are from MERRA-2). a) Difference between buoy and ice parcel (buoy - ice parcel value), and b) correlation between buoys and ice parcels (vertical dotted red line at 0 for both difference and correlation). Binwidth calculated using Freedman-Diaconis rule.**






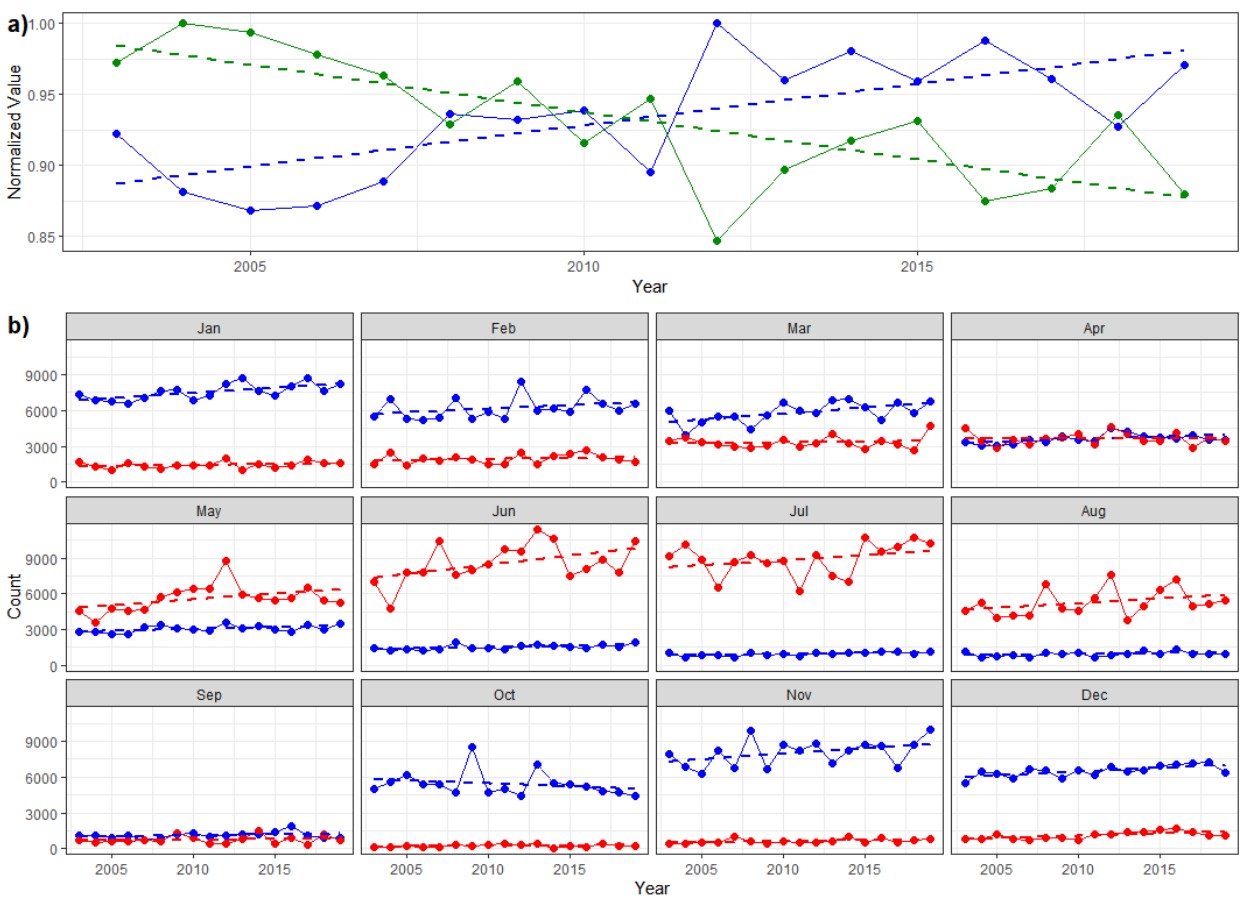

**Figure 6. Yearly summaries of sea ice parcels. a) Yearly total number of sea ice parcels (blue) and average duration of sea ice parcels (green). Both values are normalized for ease of comparison. Dotted lines are the linear fit. b) Yearly number of sea ice parcels that melt (red) and freeze (blue) by month.**




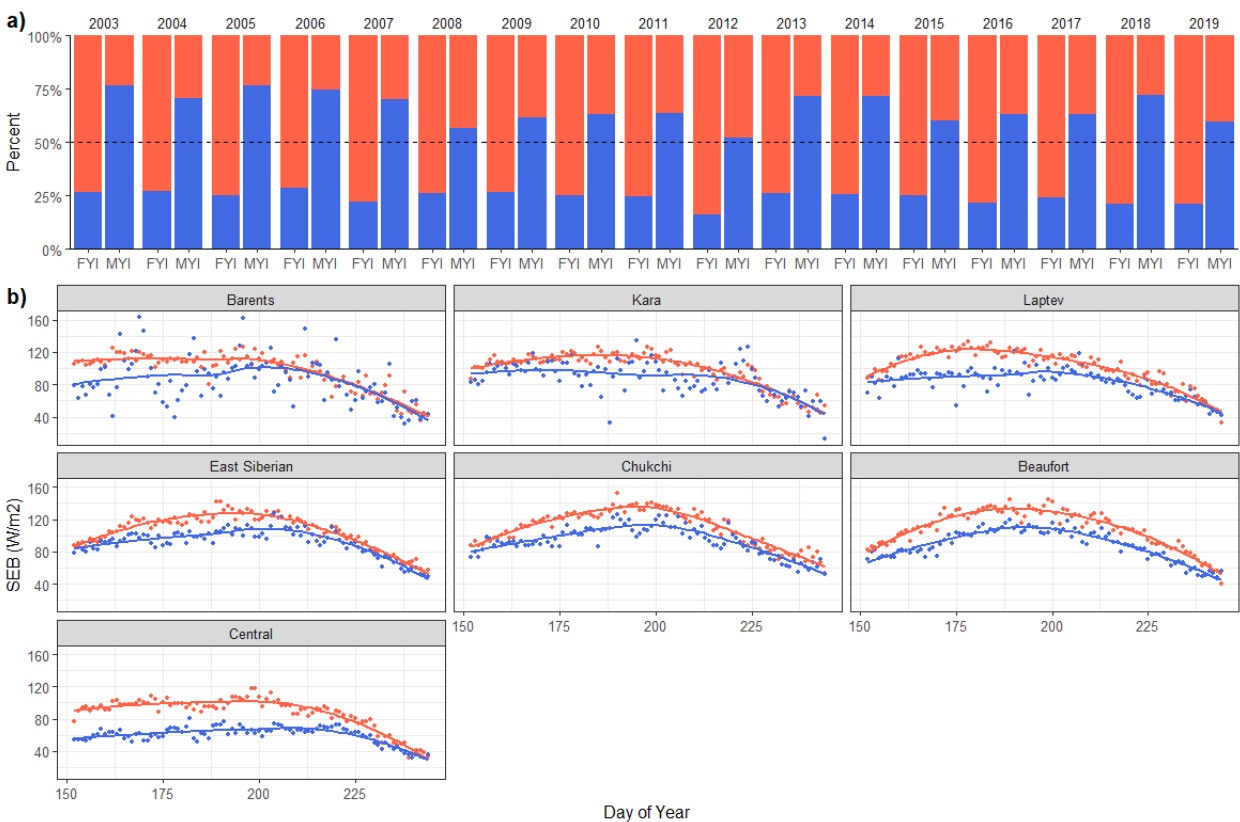

**Figure 7. Survivability of sea ice parcels. a) Percentage of first year (FYI) and multiyear (MYI) sea ice parcels that melt/survive (red/blue) the summer melt season. b) Daily averaged net SEB for June-August, grouped by region where ice parcels end. Sea ice parcels that melted out are in red, sea ice parcels that survived the melt season are in blue. Lines are the locally estimated scatterplot smoothing (LOESS) curve fit.**



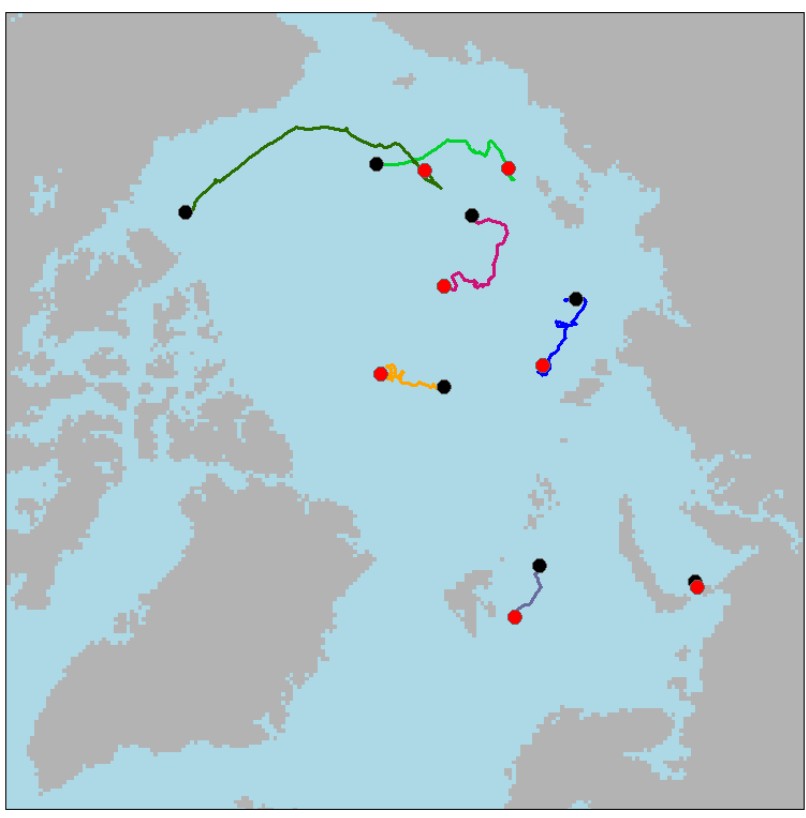

**Figure 8. Selected example parcel drift tracks from the 2017-2018 simulation. Black/red dots indicate starting/ending locations.**





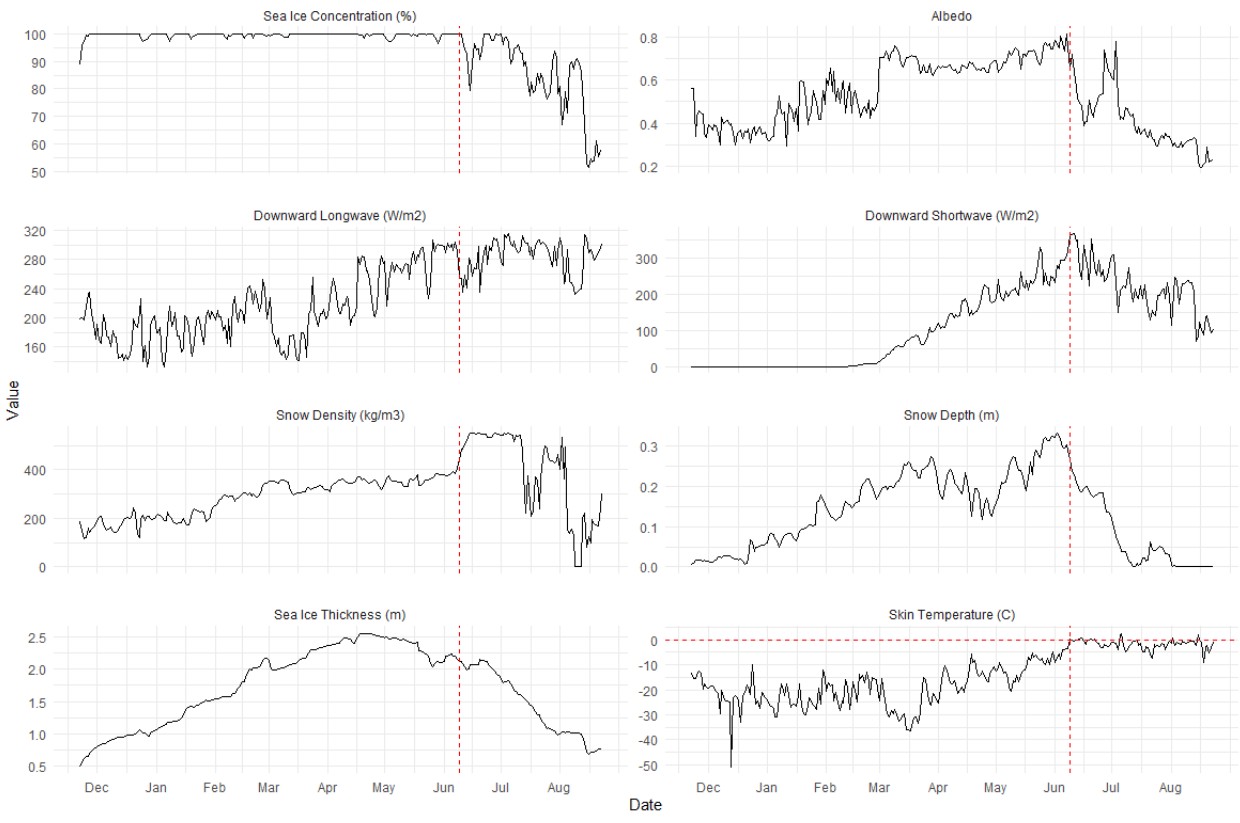

**Figure 9. Variable time-series for the light green trajectory seen in Figure 2. Horizontal dotted red line indicates 0°C. Vertical**
**dotted red lines indicate the date skin temperature first rises above the freezing point. Sea ice concentration is from the sea ice concentration CDR, sea ice thickness is from PIOMAS, snow depth and density are from SnowModel-LG forced with MERRA-2, downward longwave and shortwave radiation and albedo are from CERES, and skin temperature is from AIRS.**



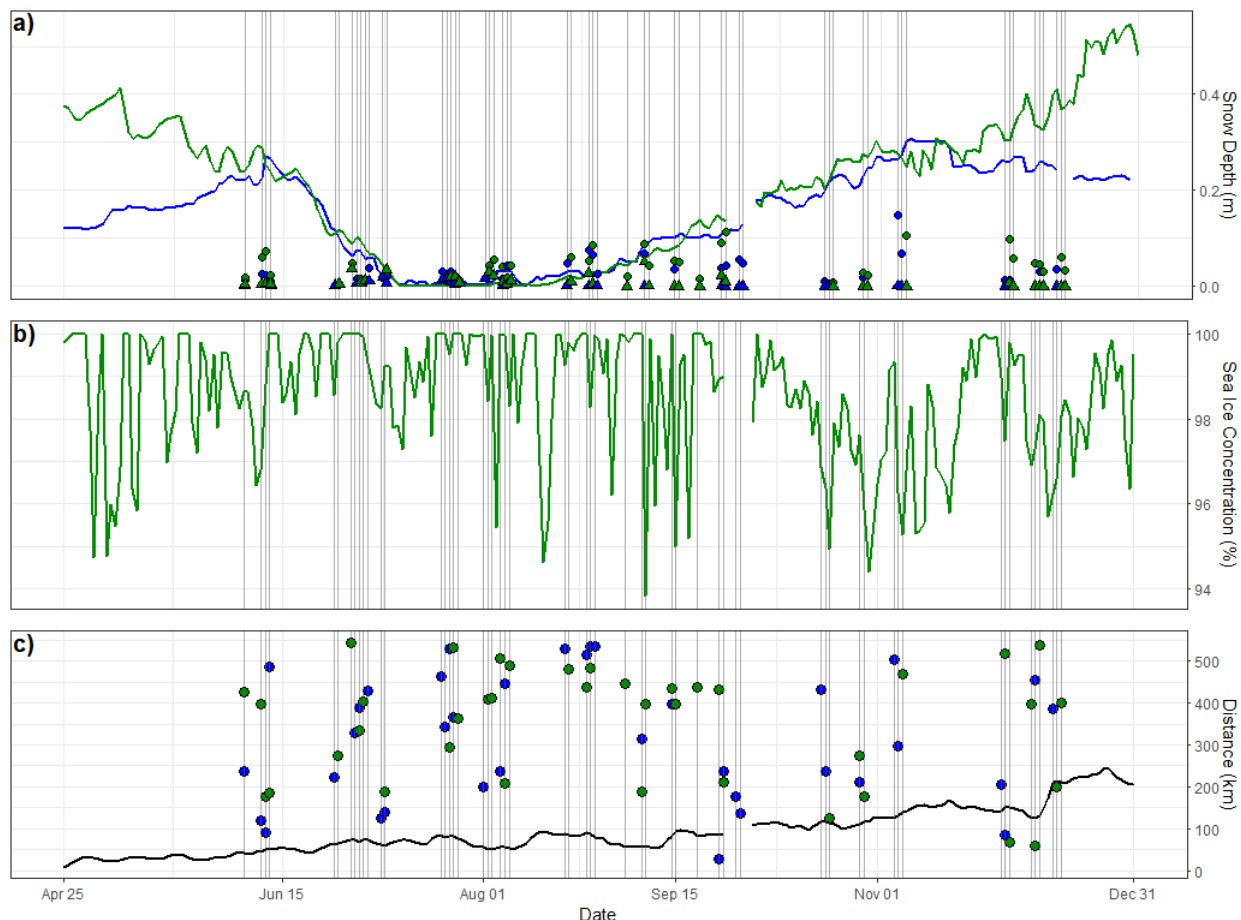

**Figure 10. Comparison of buoy ID 2004D with the nearest sea ice parcel ID 2003-2004_19893. Blue/green represents values from the buoy/sea ice parcel database. Vertical lines indicate the presence of cyclones. a) Snow depth measurements (sea ice parcel values from SnowModel-LG with MERRA2 forcing). Circles show daily total snowfall, triangles show daily total rainfall, both from the cyclone database. b) Sea ice concentration (CDR). c) Distance between the buoy/sea ice parcel and the center of the nearest cyclone. Black line indicates the distance between the buoy and sea ice parcel.**

**Table 1. The following data are used for assembling the ice parcel database.**

| Variable | Data Source | Resolution |
|---|---|---|
| **Lagrangian Sea Ice Parcel Tracking** | | |
| Sea Ice Drift | SSM/I PMW, Buoys (Tschudi et al., 2019) | Daily, 25km |
| **Sea ice conditions** | | |



| Sea Ice Concentration | SSM/I (Fetterer et al., 2017; Meier et al., 2021) | Daily, 25km |
|---|---|---|
| Sea Ice Thickness | PIOMAS (Schweiger et al., 2011) | Daily, 22km |
| Snow Depth and Density | SnowModel-LG (Liston et al., 2020) | Daily, 25km |
| **Surface Energy Budget** | | |
| Downwelling Shortwave (SW) Radiation | CERES (Rutan et al., 2015; Wielicki et al., 1996) | Daily, 20km |
| Upwelling and Downwelling SW and Longwave (LW) Clear-sky & All-sky Radiation | CERES | Daily, 20km |
| Albedo | CERES | Daily, 20km |
| Latent (LH) / Sensible (SH) Heat Flux | Derived from AIRS (Boisvert et al., 2013; 2015) | Daily, 25km |
| **Atmospheric Conditions and weather event classification and tracking** | | |
| Clouds fraction and type | CERES-MODIS/ CALIPSO-CloudSat (when possible) | Daily, 20km |
| Atmospheric pressure, temperature, specific humidity, total precipitation, snowfall, wind speed & direction | MERRA-2 (Gelaro et al., 2017) | Hourly/3-hourly, daily, 1/2° x 5/8° |
| Atmospheric pressure, temperature, specific humidity, rainfall, snowfall, wind speed & direction | ERA5 (Hersbach et al., 2018) | Hourly, daily, 1/2° x 1/2° |
| Atmospheric pressure, temperature, relative & specific humidity, skin temperature, surface air temperature, total precipitable water | AIRS (Susskind et al., 2014) | Daily, 25km |
| Cyclone identification, distance from center of cyclone, cardinal direction from center of cyclone (in degrees), minimum surface pressure | The Melbourne University cyclone tracking scheme (Simmonds et al., 2008; Webster et al., 2019) | 6-hourly, 1° x 1° |
| **Validation and assessment of remote observations and simulated processes** | | |





| Sea Ice Thickness, Snow Depth, Atmospheric Temperature and Pressure | The CRREL-Dartmouth Mass Balance Buoy Program (Perovich et al., n.d.) | 4-hourly, In Situ Point Source |
|---|---|---|