# Peer review of "Fate of sea ice in the 'New Arctic': A database of daily Lagrangian Arctic sea ice parcel drift tracks with coincident ice and atmospheric conditions"

_The Cryosphere, 2021_

## Author Comment (AC2)

**Authors' Response to Reviewer Comments**

The authors would like to thank the reviewer for their insightful feedback and constructive comments. We have addressed the concerns and/or incorporated each suggestion to strengthen our manuscript. Please note that each of the reviewer's comments, denoted in *italics*, is addressed below.

*General comments:*

*This paper presents a database consisting of a compilation of Lagrangian sea ice tracks combined with established satellite observations, model output, and reanalysis data. This comprehensive database spans from 2002 - 2019 and contains numerous sea ice properties and atmospheric variables. With the Lagrangian framework, i.e., by moving with the ice, the authors provide a useful addition to the more traditional, Eulerian datasets of sea ice and atmospheric properties. For example, the database could be used to study changes in the Arctic energy fluxes. The authors present two use cases for climatological and more process-orientated studies. They provide a detailed outlook of which datasets they plan to incorporate in the future.*

*The paper provides a detailed description of a Lagrangian database that will be useful to the sea ice community. It is well structured and easy to follow. A very positive aspect of the database is that the authors put a lot of effort into incorporating different datasets and providing a choice to the user. One central aspect I was missing was a detailed discussion of the uncertainties associated with the individual properties due to the spatial/temporal errors in the tracking. In addition, since the paper's focus is on the database, it provides limited new scientific insights about changes in the Arctic. I recommend extending the results section further with a more substantial case study and shortening the outlook.*

> **Authors' Response**:
>
> Thanks for your comments. We've added some discussion about the uncertainties that can arise due to errors in trajectory locations (see below). Additionally, we would like to include in future versions of this database more detailed uncertainty estimates.
>
> There are several hypotheses tied to this project at large which drove the formation of this dataset, but this is the preliminary manuscript meant to introduce the database and potential uses. With the recent push for open science in the community, we felt it necessary to publish methods and rationale used to create this database rather than focus on specific scientific insights as has been done in the past (Liston et al., 2020; Pfirman et al., 2019; Tschudi et al., 2010).

*Specific comments:*

*Reference to previous Lagrangian studies*

*The introduction would benefit from a more detailed discussion of previously conducted Lagrangian ice studies in the Arctic. So far, it briefly mentions Eulerian studies (L64-65). You may want to look at the following list (not complete):*

- *RGPS (RADARSAT Geophysical Processor System, e.g., https://agupubs.onlinelibrary.wiley.com/doi/full/10.1029/2000JC000469)*
- *Lagrangian Ice Tracking System ( http://icemotion.labs.nsidc.org/SITU/, https://ui.adsabs.harvard.edu/abs/2019AGUFM.C22D..03P/abstract)*

- *IceTrack (https://www.nature.com/articles/s41598-019-41456-y, https://tc.copernicus.org/articles/15/3897/2021/tc-15-3897-2021.html#&gid=1&pid=1)*
- *neXtSIM (model): https://tc.copernicus.org/articles/10/1055/2016/*

*Please add a short paragraph that describes the advances of your database compared to the previous work of other Lagrangian studies.*

**Authors' Response**:

Thanks for pointing out these previous studies. A Lagrangian approach has indeed been used in the past for the study of sea ice, though often not for the same application presented here. The most relevant study is Tschudi et al. (2010) which we mention in Section 2.1 as we build upon this study. But we do agree that mentioning other Lagrangian-based studies is worthwhile, so the following text has been added to line 67:

Studying sea ice from a Lagrangian framework has been used for tracing biogeochemical transport (Damm et al., 2018; Krumpen et al., 2019), ice volume flux (Krumpen et al., 2016; Kwok & Cunningham, 2002; Ricker et al., 2017), and for developing a numerical sea ice model (Rampal et al., 2016). Lagrangian tracking of coincident sea ice and atmospheric conditions has also been done (Pfirman et al., 2019; M. Tschudi et al., 2010), which we expand on here with higher temporal resolution (daily) and more complete atmospheric conditions including terms for calculating the SEB.

*Uncertainties of the Lagrangian drift tracks:*

*Sections 2.1 and 3.2.1 should be clarified by providing additional details on how you obtained the Lagrangian tracks and how the temporal and spatial uncertainty of the tracks translates into the uncertainty of the atmospheric and sea ice properties. Could you explain why you used the weekly ice motion product and interpolated it linearly to a daily resolution when a daily version of the Polar Pathfinder Sea ice Motion Vectors is available? Since sea ice motion varies substantially on short time scales, I recommend using the highest temporal resolution available if no other reasons speak against it. If this is not possible, please add a sentence why you used the weekly product in the manuscript.*

*Please estimate (or at least discuss) the uncertainty for the various parameters (ice thickness, air temperature, ....) introduced by a misplaced trajectory caused by the linear interpolation or errors in the Lagrangian tracking itself. For example, how much does the ice thickness / the air temperature vary if the ice parcel was located 100 km away from the trajectory? Is this spatial uncertainty the same in winter in summer? To evaluate the differences between the interpolated weekly sea ice velocities and the daily (or even sub-daily) velocities, you could use the daily PathFinder product, buoys, or SAR-derived motion field, e.g., https://resources.marine.copernicus.eu/product-detail/SEAICE_GLO_SEAICE_L4_NRT_OBSERVATIONS_011_006/INFORMATION.*

**Authors' Response**:

The accuracy of the ice motion data product and drift tracks have been assessed in previous works (Kwok et al., 1998; Meier et al., 2000; M. Tschudi et al., 2010, 2020) so was not repeated here. But we have added the following brief discussion of the findings:

Tschudi et al. (2020) found that the sea ice motion product can produce accurate tracking of parcels over time with little cumulative errors due to largely unbiased motion evaluations. Comparing drift tracks to the drift of the Surface Heat Budget of the Arctic Ocean (SHEBA) ice camp, Tschudi et al. (2010) found an error of 27 km over 293 days. Kwok et al. (1998) found ice motion errors of 5 to 12 km per day by comparing ERS-1 synthetic aperture radar (SAR) and drifting buoy motion to Lagrangian parcel tracks.

Errors in parameters due to errors in the Lagraingian tracking is an important issue and was raised by the other reviewer as well. We've compared parameter errors due to misplaced trajectories and errors due to modeling/sampling errors by interpolating common parameters from the input datasets to the true locations of the IMB buoys with the same methodology that was used for our Lagrangian tracks. For this comparison we now have parameters produced in our Lagrangian tracks database (Ldata), parameters produced with the same methodology but with the true buoy locations (Bdata), and the observed data from the buoys themselves (Bobs).

By comparing the differences between Ldata and Bdata (because they have the same input data the differences are due only to location differences) with Ldata and Bobs we get a sense of errors due to location differences. Figure 1 shows this comparison with points colored by distance between the Lagrangian track and the buoy. The results highlight three main points:

1. For air pressure, when ice parcels are large distances away from the buoy the main source of error is the distance between Lagraingian tracks and true locations as indicated by the points on the 1-to-1 line, specifically points that are a large distance away. Otherwise, when the points are not separated by large distances the main source of error is due to modeling/sampling.
2. For sea ice thickness and snow depth, inaccuracy of parcel location does contribute to parameter errors, but this is true even at small distances as indicated by the roughly linear relationship regardless of distance. This suggests that the spatial variability of these variables is so high that unless the location is exactly correct (unrealistic) some error will be present.
3. Most of the error in air temperature is due to modeling/sampling errors as indicated by the wide horizontal spread, little vertical spread, and well mixed distances.

[Figure]

Figure 1. Influence of location errors on parameter errors. Color bar shows the distance between the Lagrangian track and the buoy (on logarithmic scale for clarity). Gray dotted lines mark zero for each axis, red dotted line shows the 1-to-1 line.

*Evaluation of snow and ice thickness*

*The authors state (Section 3.2.2) that the large spatial variability of snow and ice thickness complicates the evaluation with point measurements from ice mass balance buoys. I agree, but this database would gain weight if the authors could present the ice and snow thickness data with a more detailed uncertainty analysis. There are other datasets available that could be used to evaluate the results of your tracks, e.g., satellite-derived ice thickness (CryoSAT/IceSAT) or observations from ULS (e.g., doi: 10.1002/2015JC011102) and electromagnetic induction (https://doi.org/10.5194/tc-15-2575-2021). If this is not possible, I recommend including more plausibility checks of the ice thickness and snow results, e.g., by analyzing the ice thickness time series (like in Fig. 9). For example: why is the ice thickness decreasing already as early as April? What causes the little bumps in the ice thickness time series? What role do sea ice dynamics play in the ice thickness increase? The authors should also discuss in the manuscript that the ice mass balance buoys do not consider dynamic thickness changes.*

**Authors' Response**:

PIOMAS has shown to be in good agreement with the additional sea ice thickness data you've mentioned (Laxon et al., 2013; Mu et al., 2018; Schweiger et al., 2011; Wang et al., 2016) so we did not redo these comparisons. Operation IceBridge was able to provide the necessary data for quantifying the spatial variability of sea ice thickness within each 25km by 25km grid cell.

Due to the uncertainties in all the input datasets, small scale analyses using our database may not be appropriate. The daily changes you point out in Figure 9 are a good example of this. However, the cumulative effects of atmosphere-sea ice interactions from the database can be used for studying the fate of sea ice (i.e. whether a parcel survives the summer melt season).

You make a good point that the buoys do not differentiate sea ice dynamics from thermodynamics. A more thorough study on the influence of dynamics versus thermodynamics is planned for future work with this database.

*Availability of datasets and uncertainty*

*Unfortunately, I could not find an example database for testing during the review process. Therefore, I cannot make any comments regarding the actual handling of the database. Please indicate in the revised version where the data will be publicly accessible after the acceptance of the manuscript. I could not check whether the individual data points come with an uncertainty estimate (from the data product and from the spatial missplacement) in the database. Where available, I would highly recommend including this information as it significantly improves the quality of climatological studies. For example, it would be interesting to see the uncertainty estimate in Figure 9, and especially in Figure 10.*

**Authors' Response**:

The database is currently undergoing review at NSIDC where it will be hosted. If the database doi becomes available prior to publication it will be included. Currently we state in the final paragraph that the database will be hosted at NSIDC.

As part of the review process with NSIDC we are updating information in the files including metadata where we can provide more information about the datasets used to create our database.

*Database uses:*

*The two examples (Section 3.3) are well-chosen to provide "proof of concepts" but contain limited new scientific insights. I recommend extending section 3.3.2 (Case Studies) with an example that provides more detailed insights into an Arctic process.*

**Authors' Response**:

The main goal of this manuscript is to introduce the database and potential uses as has been done in the past (Liston et al., 2020; Pfirman et al., 2019; Tschudi et al., 2010). The inclusion of the "proof of concept" examples was not meant to be exhaustive and were kept brief to avoid an unwieldy amount of information in this manuscript. However, we can expand on these two examples to convey new insights into Arctic processes.

*Future Additions*

*I found section 3.4 (Future Additions) too detailed for plans. Since changes might occur in the implementation of your plans, I would suggest cutting the subsections down to one section with a few details on the datasets you want to include and especially why you want to use them.*

**Authors' Response**:

Thank you for your suggestion. We have shortened the Future Works section to just a few details about future plans and additions.

***Technical corrections:***

*Title:*

- *The title was a bit misleading because I expected an in-depth analysis of the "fate of the New Arctic" from it. You could change the order of the words, like "A new database …. to study the fate of sea ice in the New Arctic" or remove the "fate of sea ice in the New Arctic."*

We can change the title as the editor/reviewers see fit.

*Abstract:*

- *L11: "transitioned": I think this process is still ongoing; consider using the present tense.*

Thanks, changed.

- *L19: "the database drift track": consider adding "the quality of the database was evaluated…"*

Thanks, changed.

- *L23: "less accurate": please specify*

Thanks, changed to "less accurate when compared to a point measurement"

*Introduction:*

- *Consider adding some more recent literature to your introduction, for example:*
  - *L30 (e.g. doi:10.1088/1748-9326/aae3ec, doi:10.1088/1748-9326/aade56)*

○ *L34 (e.g. IPCC 2021)*

○ *L38 (e.g. doi:10.1088/1748-9326/aae3ec)*

Thanks, these citations have been added where appropriate.

● *L32: "what happens in the Arctic …": Consider rephrasing this sentence to express the connection between the Arctic and the lower latitudes.*

Thanks. This sentence has been changed to "This is because 'what happens in the Arctic does not remain in the Arctic' but rather is connected with areas at lower latitudes (Francis & Vavrus, 2012; Vihma, 2014), Changes in the Arctic will have profound effects politically, economically, ecologically and climatologically on Earth. "

● *L39: "during this time": please specify which time you mean.*

Thanks, changed to "Since the early 2000s…"

● *L54-L60: Please consider adding a short note on sea ice dynamics and their role for sea ice thickness and extent.*

Thanks, the following has been added in L58: "and sea ice thickness is driven by thermodynamic and dynamic processes which also influence total sea ice extent."

● *L67: See specific comments. What about other studies that used Lagrangian tracking to study changes along the ice?*

Thanks, we've added some points discussing previous work that's been done using a Lagrangian framework.

● *L69: specify "characteristics."*

Thanks, changed to "thickness"

● *L70: "October 2002 and September 2019".*

Thanks, changed.

● *L70 "starts in 2002 as this is" : is there a word missing?*

Thanks. Yes, this is a typo.  "is" has been deleted.

● *L75: your database is very rich in information and helpful but does not include any ocean information what might be relevant for mass balance studies. Maybe mention this aspect either in the introduction or write a short discussion about it in section 3.4 when you talk about the use for sea ice mass balance studies.*

Thanks. In L314 we do mention that the conductive heat flux from the ocean through the sea ice is omitted in our calculation of SEB due to lack of observational data.

*Section 2.1:*

● *L82: Which version of the PathFinder are you using? Could you please indicate this?*

Thanks.  Version 4 of this dataset was used.  This has been added.

● *L82: See specific comments. Would you mind explaining why you interpolate the weekly product?*

In discussion with people at NSIDC, we were informed that the daily product is noisier and that using the weekly product has proven more accurate.  We've added a not about this in L82.

*Section 2.2:*

- *L112: "used for the Lagrangian tracking method described above": Do you mean that a 15% CDR ice concentration decides when to stop/start the tracking?*

Thanks. The 15% sea ice concentration from the NASA Team Algorithm is used to determine when to start/stop tracking a parcel.

- *L140: I suggest to remove "J." in the reference of "Stroeve et al. 2020"*

Thanks, changed.

*Section 2.3:*

- *L164-166: I suggest moving this paragraph to the beginning of the section, i.e., before 2.3.1. to increase the readability.*

Thanks, changed.

- *L172: are the errors given with +/-, or is it a bias in one direction? Would you please specify the sign?*

Thanks. These are RMS errors, so it is +/-. This has been added.

*Section 2.4:*

- *L215: You defined N-ICE2015 in L40.*

Thanks, deleted.

*Section 3.2*

- *L247: What were the criteria for a wrong location?*

Not wrong location, but erroneous data, e.g., the GPS did not properly record the location.

- *L248: Consider including all tracks in Figure 3a and highlight a few to understand better where those buoys were located.*

There are about 70 buoys used for comparison so including them all would make the figure difficult to read. This is why we only chose a select few.

- *253: Consider including an uncertainty estimate for the trajectories based on the region (either from the PathFinder product or from your analysis). If there are such differences between the regions, it would be useful to know this as a data user.*

Regional uncertainties are not included in the PathFinder user manual. In future versions of this database we would like to include a more comprehensive measure of uncertainty, both in the tracks and in the associated atmospheric parameters.

*Section 3.3:*

- *L286: I do not fully understand your conclusion on the increasing number of parcels. Did you remove the number of "surviving" parcels (MYI) from this number? Please specify this in the text. If not, does that mean that now a larger area of the Arctic is covered with sea ice? What else increases this number?*

Thanks. The point of this statement is to convey that in recent years, sea ice parcels are melting out and refreezing more often. This means we'll have more total sea ice parcels per year, but this doesn't mean that the total area is increasing. We've changed this text for clarification.

*Section 3.4:*

- *I suggest shortening those subsections to one section (see specific comments) and keeping the details for a second paper when you have implemented your plans. This also applies to the connection with MOSAiC.*

  Thanks, we've shortened the "Future Additions" section.

- *L.389: "from the Multidisciplinary"*

  Thanks, changed.

- *L392: "here. MOSAiC"*

  Thanks, changed.

*Section 4:*

- *L411: Please specify "less accurate" and state, e.g., the mean error for those buoy subsets.*

  Thanks, changed.

*Figures:*

- *General comments:*
  - *Consider adding legends to your figures. I found them hard to read with only the information given in the caption.*
  - *Make sure that x/y labels are easy to find and close to the plots.*
  - *Maps 3a, 4, 8 contain little information. Consider combining them into 1 or 2 figures or add more information.*

  Thanks. We can update figures for clarity.

*Figure 1:*

- *I do not understand why the "sea ice parcel" box is 3 times there and what "true location" means. Could you please explain?*

  Thanks. The three "sea ice parcel" boxes show the parcel on three consecutive dates, with the center box showing the location of the photo. "True Location" is the lat/lon of the image seen, "Location" is the lat/lon of the nearest sea ice parcel in our database. Text has been added to the caption for clarity.

*Figure 3:*

- *Would you please indicate in 3a (or any other map) the spatial zones you defined to sort your data into the subregions (Laptev, Central Arctic, …)?*

Yes, regional definitions have been added.

- *Consider adding a histogram with all data.*

Thanks, changed.

- *Why did you choose 25 km for the red line? In the text, it appeared that 100 km is your "uncertainty estimate".*

25km was chosen as this is the width of grid cells in the EASE2.0 projection.

- *Text in b is too small; please enlarge.*

Thanks, changed.

- *Add the number of buoys for each subregion that you used to calculate the histograms.*

Thanks, changed.

- *Maybe add a half-sentence about the purpose of the Freedman-Diaconis rule.*

Thanks, changed.

*Figure 4:*

- *Please consider combining this figure with Figure 3a or 8.*

Thanks, changed.

*Figure 5:*

- *I could not find the units for the parameters. If not done so far, please add them.*

Thanks, changed.

- *Please consider adding the standard deviation of the distributions to get an idea of the spread.*

Thanks, changed.

- *Would you please add the number of samples used to calculate the distributions?*

Thanks, changed.

- *I have trouble reading the plots in panel b due to the different bin widths. Could you consider using regular bins for those plots?*

Thanks, changed.

*Figure 7:*

- *Please move the x-label "Day of Year" to one of the plots*

Thanks, changed.

*Figure 8:*

- *In the text, you discuss this figure regarding the drift patterns (L320-322). However, I think that showing only one trajectory is not enough to display a full drift pattern. Please consider showing more trajectories or removing this part and combining the figure with one of the other maps.*

  Thanks, changed.

- *In addition, you use Figure 8 to display the track of the time series shown in Fig. 9. Is the track of the time series in Figure 10 also displayed in Figure 8, 3, or 4? If not, please include it in one of them.*

  Thanks, changed.

*Figure 9:*

- *Is this the time series of one of the green trajectories seen in Figure 2 or the light green in Figure 8? Would you please clarify this in your caption?*

  Thanks, this was a typo. It is the light green trajectory in FIgure 8.

- *Add "°C" for the skin temperature.*

  Thanks, changed.

- *If possible, please display uncertainties with those variables.*

  Thanks, we would like to include uncertainties in future versions of this data product.

- *Indicate the year in the graph or in the caption*

  Thanks, changed.

*Figure 10:*

- *A legend for a, c, would be very helpful.*

  Thanks, changed.

- *What causes the discrepancy in snow depth for the wintertime (April-June, Nov-Jan)?*

  Unclear.  It is perhaps a bias in the SnowModel-LG data product.

- *Corresponds the data gap end of September to the restart of your tracking? Consider indicating this and explain the missing time.*

  Yes, that is the data gap. We've added text indicating this.

- *If possible, please display uncertainties with those variables.*

  Thanks, we would like to include uncertainties in future versions of this data product.

*Table 1:*

- *Sea ice drift/Resolution: If you used the weekly product and interpolated it to daily, please indicate this in the table somehow (instead of "daily").*

Thanks, changed.

**References:**

Damm, E., Bauch, D., Krumpen, T., Rabe, B., Korhonen, M., Vinogradova, E., & Uhlig, C. (2018). The Transpolar Drift conveys methane from the Siberian Shelf to the central Arctic Ocean. *Scientific Reports*, *8*(1), 4515. https://doi.org/10.1038/s41598-018-22801-z

Krumpen, T., Belter, H. J., Boetius, A., Damm, E., Haas, C., Hendricks, S., Nicolaus, M., Nöthig, E.-M., Paul, S., Peeken, I., Ricker, R., & Stein, R. (2019). Arctic warming interrupts the Transpolar Drift and affects long-range transport of sea ice and ice-rafted matter. *Scientific Reports*, *9*(1), 5459. https://doi.org/10.1038/s41598-019-41456-y

Krumpen, T., Gerdes, R., Haas, C., Hendricks, S., Herber, A., Selyuzhenok, V., Smedsrud, L., & Spreen, G. (2016). Recent summer sea ice thickness surveys in Fram Strait and associated ice volume fluxes. *The Cryosphere*, *10*(2), 523–534. https://doi.org/10.5194/tc-10-523-2016

Kwok, R., & Cunningham, G. F. (2002). Seasonal ice area and volume production of the Arctic Ocean: November 1996 through April 1997. *Journal of Geophysical Research: Oceans*, *107*(C10), SHE 12-1-SHE 12-17. https://doi.org/10.1029/2000JC000469

Kwok, R., Schweiger, A., Rothrock, D. A., Pang, S., & Kottmeier, C. (1998). Sea ice motion from satellite passive microwave imagery assessed with ERS SAR and buoy motions. *Journal of Geophysical Research: Oceans*, *103*(C4), 8191–8214. https://doi.org/10.1029/97JC03334

Laxon, S. W., Giles, K. A., Ridout, A. L., Wingham, D. J., Willatt, R., Cullen, R., Kwok, R., Schweiger, A., Zhang, J., Haas, C., Hendricks, S., Krishfield, R., Kurtz, N., Farrell, S., & Davidson, M. (2013). CryoSat-2 estimates of Arctic sea ice thickness and volume. Geophysical Research Letters, 40(4), 732–737. https://doi.org/10.1002/grl.50193

Liston, G. E., Itkin, P., Stroeve, J., Tschudi, M., Stewart, J. S., Pedersen, S. H., Reinking, A. K., & Elder, K. (2020). A Lagrangian Snow-Evolution System for Sea-Ice Applications (SnowModel-LG): Part I—Model Description. *Journal of Geophysical Research. Oceans*, *125*(10). https://doi.org/10.1029/2019JC015913

Meier, W. N., Maslanik, J. A., & Fowler, C. W. (2000). Error analysis and assimilation of remotely sensed ice motion within an Arctic sea ice model. *Journal of Geophysical Research: Oceans*, *105*(C2), 3339–3356. https://doi.org/10.1029/1999JC900268

Mu, L., Losch, M., Yang, Q., Ricker, R., Losa, S. N., & Nerger, L. (2018). Arctic-Wide Sea Ice Thickness Estimates From Combining Satellite Remote Sensing Data and a Dynamic Ice-Ocean Model with Data Assimilation During the CryoSat-2 Period. *Journal of Geophysical Research: Oceans*, *123*(11), 7763–7780. https://doi.org/10.1029/2018JC014316

Pfirman, S., Campbell, G. G., Tremblay, B., Newton, R., & Meier, W. (2019). *Lagrangian Ice Tracking System: LITS Expanded with Arctic and Antarctic Environmental Data*. *2019*, C22D-03.

Rampal, P., Bouillon, S., Ólason, E., & Morlighem, M. (2016). neXtSIM: A new Lagrangian sea ice model. *The Cryosphere*, *10*(3), 1055–1073. https://doi.org/10.5194/tc-10-1055-2016

Ricker, R., Hendricks, S., Girard-Ardhuin, F., Kaleschke, L., Lique, C., Tian-Kunze, X., Nicolaus, M., & Krumpen, T. (2017). Satellite-observed drop of Arctic sea ice growth in winter 2015–2016. *Geophysical Research Letters*, *44*(7), 3236–3245. https://doi.org/10.1002/2016GL072244

Schweiger, A., Lindsay, R., Zhang, J., Steele, M., Stern, H., & Kwok, R. (2011). Uncertainty in modeled Arctic sea ice volume. *Journal of Geophysical Research: Oceans*, *116*(C8). https://doi.org/10.1029/2011JC007084

Tschudi, M., Fowler, C., Maslanik, J., & Stroeve, J. (2010). Tracking the Movement and Changing Surface Characteristics of Arctic Sea Ice. *IEEE Journal of Selected Topics in Applied Earth Observations and Remote Sensing*, *3*(4), 536–540. https://doi.org/10.1109/JSTARS.2010.2048305

Tschudi, M., Meier, W. N., & Stewart, J. S. (2020). An enhancement to sea ice motion and age products at the National Snow and Ice Data Center (NSIDC). *The Cryosphere*, *14*(5), 1519–1536. https://doi.org/10.5194/tc-14-1519-2020

Wang, X., Key, J., Kwok, R., & Zhang, J. (2016). Comparison of Arctic Sea Ice Thickness from Satellites, Aircraft, and

PIOMAS Data. *Remote Sensing*, *8*(9), 713. https://doi.org/10.3390/rs8090713

---

## Author Comment (AC3)

**Authors' Response to Reviewer Comments**

The authors would like to thank the reviewers for their insightful feedback and constructive comments. We have addressed the concerns and/or incorporated each suggestion to strengthen our manuscript. Please note that each of the reviewer's comments, denoted in *italics*, is addressed below.

*General comments:*

*This paper documents the assembly of a database of lagrangian drift tracks that includes many of the variables needed to study changes in the surface energy budget (SEB).*

*They assert that the database can be used to study a range of SEB processes, and present some results from the assembled dataset. However, the authors also show that the drift tracks have mean errors of 82.6 km, and can be as high as 500 km in certain areas, so the "tracks" are probably not actually following the same parcel of sea ice.*

**Authors' Response**:

Thanks for your comments. The accuracy of the Lagrangian tracks certainly plays a role in the accuracy and usefulness of this database. We've added some discussion points addressing this (see below) and still believe this database to be useful for large scale analyses. While the day to day changes in parameter values may not be realistic, the long term or cumulative effects of these parameters on a sea ice parcel can be used for subseasonal to seasonal analyses.

*Detailed Major Suggestions and Comments:*

1. *The paper needs a good scientific hypothesis or question to guide the research.*

   *This paper jumps into the middle of the scientific process by first producing a database, then trying to find questions that the database may help answer. A more fruitful approach would be to find a scientific hypothesis, then assemble data to test that hypothesis.*

**Authors' Response**:

There are several hypotheses tied to this project at large which drove the formation of this dataset, but this is the preliminary manuscript meant to introduce the database and potential uses. With the recent push for open science in the community, we felt it necessary to publish methods and rationale used to create this database rather than focus on specific scientific insights as has been done in the past (Liston et al., 2020; Pfirman et al., 2019; Tschudi et al., 2010).

2. *Mean errors of 82.6 km for the lagrangian tracks may or may not be large depending on the scientific question a person is trying to answer.*
   a. *For example, if one were trying to understand the roll of large scale cyclones, then this probably is not an issue, but if one were trying to understand the small scale changes in ice concentration, then this error is unacceptable. Looking at Figure 1, the authors mark a 25km x 25 km box. A shift of even just a few km, shows that we are looking at an area of much higher sea ice concentration than the sea ice parcel that is highlighted. The SEB in the marked pixel is much different that the SEB in the parcels surrounding this.*
   b. *If the mean errors are this large in reproducing the tracks of a buoy that is included in the gridded ice motion database, how much larger are the errors in areas without buoys?*
   c. *Given the errors in reproducing lagrangian tracks, why not just use the actual drift tracks of the buoys? For example, the Ice Mass Balance buoys measure many of the quantities assembled here.*

> d. *Section 3.2.2, and Figure 5: Are the differences seen in each of the panels due to real physical changes in the parcel compared to the buoy observations, or due to errors in the lagrangian tracks?*

**Authors' Response**:

a) Thanks, yes this database is better suited to large scale analyses rather than fine temporal/spatial analyses. Many of the hypotheses we wish to address in future work are concerned with longer term (weekly/seasonal) influences of atmospheric parameters on sea ice. In this regard, the database captures summary statistics well. Table 1 below shows the average difference of maximum, mean, and minimum values of parameters between individual buoys and the associated Lagrangian sea ice parcel. We've added some discussion of this in Section 3.2.1.

Table 1. Average differences of the maximum, mean, and minimum value of parameters between IMB buoys and Lagrangian parcels.

| Variable | Maximum | Mean | Minimum |
|---|---|---|---|
| AirPressureDiff | 1.17 | -0.07 | -1.60 |
| AirTempDiff | 0.74 | -0.88 | -2.80 |
| IceThicknessDiff | -0.33 | -0.01 | 0.45 |
| SnowDepthDiff | -0.06 | -0.02 | 0.01 |

b) Conducting a detailed validation of trajectory accuracy is difficult because most observations of sea ice drift (for example, buoys) are included in the sea ice motion product. However, Tschudi et al. (2010) compared the drift of the Surface Heat Budget of the Arctic Ocean (SHEBA) ice camp with sea ice parcel tracks using this same Lagrangian method and found an error of 27 km over 293 days.

c) The benefit of creating this database using Lagrangian tracks rather than buoy drift tracks is that we can create a much larger database. Our database contains over 1 million tracks. However, creating a database using buoys as you've suggested could be a nice supplement to the database we've created and could be done in the future.

d) The simple answer is both, but we've now done an analysis of parameter errors caused by errors in location. We've compared parameter errors due to misplaced trajectories and errors due to modeling/sampling errors by interpolating common parameters from the input datasets to the true locations of the IMB buoys with the same methodology that was used for our Lagrangian tracks. For this comparison we now have parameters produced in our Lagrangian tracks database (Ldata), parameters produced with the same methodology but with the true buoy locations (Bdata), and the observed data from the buoys themselves (Bobs).

By comparing the differences between Ldata and Bdata (because they have the same input data the differences are due only to location differences) with Ldata and Bobs we get a sense of errors due to

location differences. Figure 1 shows this comparison with points colored by distance between the Lagrangian track and the buoy. The results highlight three main points:

1. For air pressure, when ice parcels are large distances away from the buoy the main source of error is the distance between Lagraingian tracks and true locations as indicated by the points on the 1-to-1 line, specifically points that are a large distance away. Otherwise, when the points are not separated by large distances the main source of error is due to modeling/sampling.
2. For sea ice thickness and snow depth, inaccuracy of parcel location does contribute to parameter errors, but this is true even at small distances as indicated by the roughly linear relationship regardless of distance. This suggests that the spatial variability of these variables is so high that unless the location is exactly correct (unrealistic) some error will be present.
3. Most of the error in air temperature is due to modeling/sampling errors as indicated by the wide horizontal spread, little vertical spread, and well mixed distances.

[Figure]

Figure 1. Influence of location errors on parameter errors. Color bar shows the distance between the Lagrangian track and the buoy (on logarithmic scale for clarity). Gray dotted lines mark zero for each axis, red dotted line shows the 1-to-1 line.

3. *All the different datasets assembled here also have their own errors. As with comment 2 above, whether these errors are acceptable depends on the scientific questions we are trying to answer.*
    a. *One thing to note is that a "lagrangian approach" may also be taken by directly using many of the disparate datasets they assembled here. For example, PIOMAS includes many of these variables as forcing or as estimates from the model. PIOMAS is well documented so the errors,*

*biases and uncertainties are known. The model can give us a "self consistent" framework to do lagrangian studies by tracking a parcel using the ice motion provided by the model.*

b. *By assembling disparate datasets as is done here, we lose the "self consistency" of each data set and quantifying the errors in our results becomes difficult. Following example, looking at figure 9, the sea ice thickness obtained from PIOMAS starts declining in May long before the onset of melt derived from AIRS skin temperatures. How can we explain this given the variables assembled?*

c. *Sea ice thickness also increases in PIOMAS just before the onset of melt in June (Fig. 9). What forces this change? Or is there simply a shift in the pixels that they are tracking?*

d. *A more thorough discussion of errors for each dataset should be included in section 2*

**Authors' Response**:

Thanks for pointing this out, this is a good point. Due to the uncertainties in all the input datasets, small scale analyses using our database may not be appropriate. The daily changes you point out in Figure 9 are a good example of this. However, the cumulative effects of atmosphere-sea ice interactions from the database can be used for studying the fate of sea ice (i.e. whether a parcel survives the summer melt season) as mentioned earlier (response to comment 2a, Table 1).

You make a good point that we are incorporating errors from each data source into this database, but using multiple data sources for the study of sea ice is often done. We've included many atmospheric and sea ice/snow variables in the hope of a wide range of uses of this database, but all variables do not necessarily need to be used in conjunction.

4. *Reading through their abstract and conclusions, the primary contributions of this paper to science are: 1) they produced a lagrangian data base, and 2), they find an increase in the number of sea ice parcels over time. Both these findings are moot given that they may not be tracking the same parcel of sea ice, and since they note that their lagrangian drift tracks are significantly slower near Fram Strait where most parcels of sea ice is exported from the Arctic. The increase in sea ice parcels over time can probably be attributed to more of their parcels "surviving" since less are exported through Fram Strait compared to the real world.*

**Authors' Response**:

Thanks for your comment. A third major contribution is including terms for calculating the surface energy budget, the usefulness of which is demonstrated in Section 3.3.1. The increase in the number of sea ice parcels over time is likely due to increased variability in sea ice conditions, in particular sea ice concentration. As the generation/termination of sea ice parcels is determined by the 15% concentration threshold, counting the number of parcels that are generated/terminated is essentially a quantification of the variability of low sea ice concentrations.

As mentioned above, the aim of this manuscript is to introduce this database and to show some examples of its use. With the recent push for open source scientific research within the community, publications of this nature are not unheard of (Liston et al., 2020; Pfirman et al., 2019; Tschudi et al., 2010). There are several hypotheses tied to the funding of this project which drove the creation of this database and will be addressed in future work.

***Minor suggestions and comments:***

*Line 35: Change "known as" to "attributed to".*

Thanks, changed.

*Figure 5: Add units to each row of plots.*

> Thanks, changed.

*Figure 7a: separate FYI and MYI bars so that we may be able to see any differences or trends from year to year. Interspersing FYI and MYI as shown makes it hard to see things.*

> Thanks, changed.

*Figure 9: Mark cyclones as in Fig. 10. It would be interesting to see if cyclones are related to the changes in in snow depth, or sea ice thickness shown here.*

> Thanks, changed.

**References:**

Liston, G. E., Itkin, P., Stroeve, J., Tschudi, M., Stewart, J. S., Pedersen, S. H., Reinking, A. K., & Elder, K.

(2020). A Lagrangian Snow-Evolution System for Sea-Ice Applications (SnowModel-LG): Part I—Model

Description. *Journal of Geophysical Research. Oceans*, *125*(10). https://doi.org/10.1029/2019JC015913

Pfirman, S., Campbell, G. G., Tremblay, B., Newton, R., & Meier, W. (2019). *Lagrangian Ice Tracking*

*System: LITS Expanded with Arctic and Antarctic Environmental Data*. *2019*, C22D-03.

Tschudi, M., Fowler, C., Maslanik, J., & Stroeve, J. (2010). Tracking the Movement and Changing Surface

Characteristics of Arctic Sea Ice. *IEEE Journal of Selected Topics in Applied Earth Observations and Remote*

*Sensing*, *3*(4), 536–540. https://doi.org/10.1109/JSTARS.2010.2048305